# The Road towards Gene Therapy for X-Linked Juvenile Retinoschisis: A Systematic Review of Preclinical Gene Therapy in Cell-Based and Rodent Models of XLRS

**DOI:** 10.3390/ijms25021267

**Published:** 2024-01-19

**Authors:** Isa van der Veen, Andrea Heredero Berzal, Céline Koster, Anneloor L. M. A. ten Asbroek, Arthur A. Bergen, Camiel J. F. Boon

**Affiliations:** 1Department of Ophthalmology, Amsterdam UMC, University of Amsterdam, Meibergdreef 9, 1105 AZ Amsterdam, The Netherlands; i.vanderveen@amsterdamumc.nl (I.v.d.V.); a.herederoberzal@amsterdamumc.nl (A.H.B.); c.koster@amsterdamumc.nl (C.K.); aabergen@amsterdamumc.nl (A.A.B.); 2Department of Human Genetics, Amsterdam UMC, University of Amsterdam, Meibergdreef 9, 1105 AZ Amsterdam, The Netherlands; a.l.tenasbroek@amsterdamumc.nl; 3Department of Ophthalmology, Leiden University Medical Center, Leiden University, Albinusdreef 2, 2333 ZA Leiden, The Netherlands

**Keywords:** systematic review, meta-analysis, X-linked juvenile retinoschisis (XLRS), retinoschisin (RS1), adeno-associated viral vector (AAV), non-viral, gene therapy

## Abstract

X-linked juvenile retinoschisis (XLRS) is an early-onset progressive inherited retinopathy affecting males. It is characterized by abnormalities in the macula, with formation of cystoid retinal cavities, frequently accompanied by splitting of the retinal layers, impaired synaptic transmission of visual signals, and associated loss of visual acuity. XLRS is caused by loss-of-function mutations in the retinoschisin gene located on the X chromosome (*RS1*, MIM 30083). While proof-of-concept studies for gene augmentation therapy have been promising in in vitro and rodent models, clinical trials in XLRS patients have not been successful thus far. We performed a systematic literature investigation using search strings related to XLRS and gene therapy in in vivo and in vitro models. Three rounds of screening (title/abstract, full text and qualitative) were performed by two independent reviewers until consensus was reached. Characteristics related to study design and intervention were extracted from all studies. Results were divided into studies using (1) viral and (2) non-viral therapies. All in vivo rodent studies that used viral vectors were assessed for quality and risk of bias using the SYRCLE’s risk-of-bias tool. Studies using alternative and non-viral delivery techniques, either in vivo or in vitro, were extracted and reviewed qualitatively, given the diverse and dispersed nature of the information. For in-depth analysis of in vivo studies using viral vectors, outcome data for optical coherence tomography (OCT), immunohistopathology and electroretinography (ERG) were extracted. Meta-analyses were performed on the effect of recombinant adeno-associated viral vector (AAV)-mediated gene augmentation therapies on a- and b-wave amplitude as well as the ratio between b- and a-wave amplitudes (b/a-ratio) extracted from ERG data. Subgroup analyses and meta-regression were performed for model, dose, age at injection, follow-up time point and delivery method. Between-study heterogeneity was assessed with a Chi-square test of homogeneity (I^2^). We identified 25 studies that target RS1 and met our search string. A total of 19 of these studies reported rodent viral methods in vivo. Six of the 25 studies used non-viral or alternative delivery methods, either in vitro or in vivo. Of these, five studies described non-viral methods and one study described an alternative delivery method. The 19 aforementioned in vivo studies were assessed for risk of bias and quality assessments and showed inconsistency in reporting. This resulted in an unclear risk of bias in most included studies. All 19 studies used AAVs to deliver intact human or murine *RS1* in rodent models for XLRS. Meta-analyses of a-wave amplitude, b-wave amplitude, and b/a-ratio showed that, overall, AAV-mediated gene augmentation therapy significantly ameliorated the disease phenotype on these parameters. Subgroup analyses and meta-regression showed significant correlations between b-wave amplitude effect size and dose, although between-study heterogeneity was high. This systematic review reiterates the high potential for gene therapy in XLRS, while highlighting the importance of careful preclinical study design and reporting. The establishment of a systematic approach in these studies is essential to effectively translate this knowledge into novel and improved treatment alternatives.

## 1. Introduction

### 1.1. Clinical Features of Retinoschisis

Sight is a fundamental driver of daily function. Vision loss or impairment impacts affected individuals in all aspects of life, resulting in a loss of independence and a significantly lower quality of life [1]. In the vertebrate eye, light is converted to electrical signals in the retina, from where visual information is projected to sensory areas of the brain. In many retinal degenerative diseases, retinal signaling disruptions translate directly into vision loss. A range of genetic components determines the development and maintenance of retinal cells. Mutations in these genes can result in inherited retinal diseases (IRDs). X-linked juvenile retinoschisis (XLRS, MIM 312700) is a relatively common IRD with an estimated prevalence of 1:5000–1:25,000. This disease was first described in 1898 by ophthalmologist Josef Haas, who reported the condition in two young brothers [2]. XLRS mainly affects boys in the first decade of life, although severe cases have been diagnosed as young as three months old. However, patients are typically diagnosed at school age [3,4], as then moderate loss of visual acuity (VA) is easily noticed by visual screening programs or because children experience difficulties with reading. Due to its X-linked recessive inheritance pattern, only a limited number of female cases of XLRS have been reported. Nevertheless, homozygous females have shown a XLRS phenotype, and heterozygous females can very rarely also show XLRS features as a consequence of X-chromosome inactivation [3,5]. Female carriers are generally asymptomatic, although they can present with peripheral retinoschisis in rare cases [6,7].

XLRS disease severity and progression are highly variable, even among family members [4]. In most patients, the loss of VA progresses slowly in the first decades of life, with up to 25% of patients becoming blind by the age of 60 due to the progressive development of macular atrophy, along with retinal pigmented epithelium (RPE) dysfunction and macular pigmentary changes [4,8]. Around 5–20% of patients experience complications in the form of rhegmatogenous retinal detachments and/or recurrent vitreous hemorrhages, often already at a young age [3,4,9]. Other rare complications include glaucoma and optic disc pallor [4].

The most common clinical finding in XLRS patients is bilateral macular schisis, which occurs in virtually all patients (Figure 1D). This can be identified using fundus photographs, but optical coherence tomography (OCT) scanning is generally used to visualize pathological features in a highly detailed manner. On fundus examination, a spoke-wheel pattern of microcystic changes radiating outwards from the foveola is typically seen [10] (Figure 1D). The schisis is seen in both the nuclear and plexiform layers of the retina and can involve the inner, middle and outer retinal layers [4,11]. With time, these cystoid schisis lesions can gradually diminish, leaving atrophic changes in the macula that may be mistaken for other inherited or non-inherited macular diseases [4]. Approximately 50% of affected individuals also have peripheral retinoschisis, and have an increased risk of developing complications such as retinal detachment and/or vitreous hemorrhage [4,9]. Other morphological retinal findings include exudative retinopathy, vitreous veils and retinal lattice-like lesions [4].

On functional evaluation, the amplitude of the dark-adapted b-wave on an electroretinogram (ERG) is disproportionally reduced, sometimes to the extent that the ratio between the b- and a-wave amplitudes (b/a-ratio) is below 1, and the b-wave does not cross the baseline, resulting in an ‘electronegative’ waveform [4,12]. The a-wave on an ERG is believed to represent photoreceptor (PR) function, whereas the b-wave is indicative of ON bipolar cell activity. An electronegative ERG is thought to imply a deficiency of synaptic signal propagation between the PRs and the bipolar cells [13].

### 1.2. The Retinoschisin Gene and Its Expression

Sauer and colleagues identified mutations in retinoschisin (*RS1*) (OMIM#312700) as the cause of XLRS in 1997 [14]. *RS1* is located in the chromosomal region Xp22.1 and has a genomic size of 32,421 kb. The gene contains six exons, and is transcribed into a 3.1 kb mRNA which translates into a 224-amino-acid protein [14,15] (Figure 1A). More than 200 *RS1* mutations have thus far been associated with XLRS, including point mutations, splice-site mutations, deletions and insertions [4,16]. Missense mutations are the most prevalent, with the majority located between exons 4 and 6 [4,17]. *RS1* is mainly expressed by the photoreceptors and bipolar cells in the mature retina [11,18]. *RS1* is also expressed in the pinealocytes in the pineal gland, and can be found in low levels in the brain, lung and thyroid, where the role of the protein remains to be elucidated [19,20].

### 1.3. Retinoschisin Protein and Function

The active biological conformation of the protein RS1 is a homo-oligomeric octamer that is secreted from both photoreceptors and bipolar cells [14,21,22,23]. Each RS1 subunit consists of four distinct domains: the N-terminal signal sequence, encoded by exons 1 and 2, the unique RS1 domain encoded by exon 3, the discoidin domain encoded by exons 4--6, and the C-terminal segment (Figure 1B). The discoidin domain mediates communication between cells and tissues [24] and is shared by several well-conserved extracellular cell surface proteins [25,26,27,28,29,30,31]. The function of the discoidin domain in RS1 is thought to be crucially involved in the maintenance of retinal architecture as well as synaptic connectivity between layers [11,32].

#### Localization

During development, RS1 (protein and mRNA) is expressed in all retinal neurons with exception of horizontal cells [33]. In mice, the protein is first observed around postnatal day six (P6), and adult expression patterning is reached at P12 [18]. In the mature retina only PRs and bipolar cells are involved in RS1 synthesis, subsequently, in its secretion [18,33,34]. As a consequence, RS1 is detected throughout the inner and outer mature retina as a secreted protein ([18,35]. In human, bovine and murine, RS1 protein is detected in high levels in the inner segment (IS) layer, particularly in the ellipsoid zone adjacent to the outer segments (OS) [18]. It is also present in the inner nuclear layer (INL) and to a lesser degree in the outer nuclear layer (ONL) and outer plexiform layer [18]. No RS1 protein is found in retinal ganglion cells or within the inner limiting membrane (ILM) [18].

### 1.4. Treatment Options

Currently, there is no effective treatment for XLRS. Attempts to treat this disease have historically been focused on reducing the schisis cavities. Invasive techniques like vitreoretinal surgery are only employed for patients experiencing severe complications like retinal detachments or significant epiretinal membrane of the macula [36]. Carbonic anhydrase inhibitors, such as dorzolamide, have been used to treat XLRS patients, by reducing the intraretinal schisis fluid cavities [37,38]. Carbonic anhydrase inhibitors work by inhibiting the carbonic anhydrase receptors on the membrane of the retinal neurons and the RPE. This leads to an acidification of the subretinal space and, subsequently, an increased transport of fluid across the RPE, which promotes adhesion of the retinal layers [39]. These medications may achieve a variable amount of reduction in schisis cavities in XLRS, but without marked improvements in VA [38,39,40,41,42,43].

Despite the initial promise of carbonic anhydrase inhibitors as a potential symptomatic treatment, the lack of reproducible improvements in VA has led to a shift in focus in the (re)search for XLRS treatment. A main target has become to address the actual underlying cause, the mutations in the *RS1* gene, and the fact that XLRS appears to be an attractive target for gene therapy, being a monogenic disease with a clear and relatively uniform clinical phenotype [4].

#### 1.4.1. Adeno-Associated Viral Vector-Mediated Gene Augmentation Therapy

Recent advances in genetic engineering technology have allowed the development of gene therapy methods based on adeno-associated viral vectors (AAVs). AAVs are a popular vehicle choice and are well-established vectors for preclinical and clinical gene delivery. This is due to the recent advances in AAV capsid research, their high transduction efficiency, high biosafety, reduced immunogenicity and their long-term and stable genome expression [44,45,46]. Nevertheless, AAVs also encounter some challenges, such as their high manufacturing costs, their small packaging capacity (less than 4.7 kb), and potential risks such as inflammation and retinal atrophy in the case of ocular use [44,47]. Despite these limitations, AAVs were approved by the Food and Drug administration (FDA) and European Medicine Agency (EMA) to treat *RPE65*-associated Leber’s congenital amaurosis, and are currently used in ongoing clinical trial for a broad range of inherited retinal diseases [48,49,50,51]. Different AAV serotypes have distinct cellular tropism and transduction capacities, depending on their interaction with membrane receptors and cellular mechanisms for uptake and trafficking [52]. Within the AAV genome, the promoter is an essential DNA regulatory element that determines where and to what degree the transgene is expressed [53]. Therefore, the promoter’s choice and the serotype are both essential to the design to achieve successful gene augmentation.

Gene delivery for treating XLRS should be focused on the outer retina, specifically the photoreceptors, and in the inner retina targeting bipolar cells, since they are involved in RS1 synthesis in the mature retina [18,33,34]. Transduction of *RS1* in photoreceptors of a Rs1h-deficient mouse model showed significant improvements in RS1 production and secretion and a reduced the number of cavities, improved retinal organization and led to functional rescue [54]. In an effort to identify new potential cellular targets, transduction in the Rs1h-deficient mouse model was extended to Müller cells, given their implication in RS1 trafficking, their distribution throughout the retina, and their preservation even in late stages of the disease [54]. Despite the favorable outcomes in terms of RS1 production and secretion in Müller cells, this approach did not show reduced retinoschisis in this model [54]. This underscores the complex nature of RS1 as a secreted protein in the extracellular space, and highlights that delivery to just any retinal cell type is insufficient to reduce pathogenic hallmarks.

Extensive research has been conducted regarding safety concerns with regard to the use of AAVs. Nevertheless, and despite the eye’s relative immune privilege, immune responses to high AAV doses are still reported in in vivo studies [55,56]. To overcome this, non-viral methods have emerged as a potentially attractive alternative to conventional viral delivery methods.

#### 1.4.2. Non-Viral Gene Augmentation or Gene Editing

Non-viral vectors encapsulate and transport DNA, mRNA, small interfering RNA (siRNA) and microRNA (miRNA) [57]. Regardless of the construct of interest, non-viral vectors will protect them from degradation, facilitating their transportation [58]. These delivery systems can also be used to perform gene therapy, via gene integration, gene augmentation and Clustered Regularly Interspaced Short Palindromic Repeats (CRISPR)/Cas9, which know less safety concerns [59]. In addition, they allow antisense oligonucleotide introduction, an increasing avenue of interest to treat IRDs [57,60]. Non-viral methods can be classified as chemical, relying on the capacity of positive lipids, peptides or polymers to introduce the negatively charged DNA of interest into cells [61] or physical, by using temperature or electrical shock to achieve nucleic acid integration [62]. They have become attractive due to more packaging capacity compared to AAVs, the feasibility of producing large amounts of non-viral particles reducing production costs, the straightforward option of modification, and their long-viral stability [63,64,65]. These non-viral vectors appear to have less immunotoxicity, in contrast to AAV or related viral vectors potentially [66]. Despite these advantages, non-viral methods have several limitations, such as treatment duration and transduction efficiency [67].

XLRS seems well suited for gene therapy, due to its monogenic character, relatively early-onset, slowly progressive phenotype, and the non-proliferative nature of retinal cells [4]. In fact, significant results with several AAV-mediated gene augmentation therapies were obtained in rodent models, which will be explored in this systematic review. However, a recent clinical trial using recombinant AAV-based gene augmentation therapy AAV8-scRS/IRBPhRS (scRS = self-complementary vector genome containing a modified human retinoschisin promoter; IRBP = interphotoreceptor retinoid-binding protein enhancer) in nine patients failed to improve symptoms in all but one (ClinicalTrial.gov: NCT02317887) [68]. Whether this can be attributed to the limited translational value of the models used or inadequate (pre)clinical study design remains to be elucidated. This review aims to systematically summarize and evaluate the existing in vivo and in vitro studies that target RS1 deficiency through viral and non-viral gene augmentation or gene editing therapy to improve (pre-)clinical study designs.

## 2. Methods

### 2.1. Literature Search

A systematic review protocol was completed before starting any data extraction (Appendix A). Online databases MEDLINE (through PubMed), EMBASE and Web of Science were used to identify original research articles published before May 2021. The search was repeated to include new results up to October 2023. There was no restriction on language or publication date. A combination of terms for retinoschisis, genetic therapy, retina, animal, induced pluripotent stem cells and organoids was used. A comprehensive overview of the search strategy is included in the Appendix A (Appendix A). Conference abstract screening and Biorxiv database were used to find the grey literature, selecting only relevant papers for the research question. The snowballing method was used to recursively track possible additional relevant studies cited in the found results. This did not result in new studies to be included.

### 2.2. Eligibility and Exclusion Criteria

Preclinical studies that used rodent models or in vitro models to study gene therapies (gene augmentation and/or gene editing) targeting XLRS were included. Ex vivo studies, where the virus was not directly delivered into the animal, were excluded. Eligibility screening was conducted according to predefined exclusion criteria, as outlined in the protocol in the Appendix A, by two independent reviewers (I.V. and A.H.B.) in two phases: (1) title and abstract screening and (2) full text screening. In the first phase, results were evaluated based on title and abstract, excluding papers based on the following reasons: not related to XLRS, no gene therapy, no intervention, no original data, unsuitable study design, wrong model, wrong publication type or wrong study duration. In the case of discrepancies in judgment, they were discussed until a unanimous verdict was reached. The studies were further evaluated in the second round by full-text screening, and papers were excluded based on the following: no full text available (e.g., conference abstracts) and commentary on included paper. If no full text was available online, the authors were contacted via email to obtain the full text. All eligible studies were in English. All included studies were assigned a study ID formatted as First-Author Publication year. A third screening round was performed to select eligible articles for meta-analyses. Outcome parameters selected for meta-analysis were a-wave amplitude, b-wave amplitude, and b/a-ratio on ERG. Outcome parameters not eligible for meta-analysis, and therefore selected for qualitative analysis, were scanning laser ophthalmoscopy (SLO), OCT, vision-based behavior, and immunohistochemistry (IHC). Multiple experimental groups described in one reference were distinguished from one another by labeling them a through z.

### 2.3. Data Extraction

The following data were collected from all articles: the first author name, cell line/animal model that was used in the study, mutation, delivery method (viral or non-viral), control group, follow-up timepoints, and outcome measurements. This initial data extraction allowed the following classification: (1) AAV-mediated gene augmentation therapy and (2) alternative and non-viral methods that target *RS1*. For (1) AAV-mediated gene augmentation studies, the following data were collected: genetic background, intervention time point, injection type, injection volume, needle size, total virus genomes per eye, control group type, age at intervention, and follow-up timepoints. For (2) studies using alternative and non-viral methods to target *RS1*, the following data were collected: delivery method, formulations, vector size, intervention timepoints, and follow-up timepoints. For meta-analysis, outcome data were extracted from text, tables or from graphs by digital screen ruler (WebPlotDigitizer, version 4.6).

### 2.4. Risk-of-Bias Assessment

To assess for possible bias in the included in vivo studies, the SYRCLE tool for bias assessment in animal studies was used [69]. All rodent papers, excluding rabbits, were scored by two independent reviewers (I.V and A.H.B). In the case of discrepancies, they were discussed until unanimity was reached. First, the reporting of four key quality indicators was evaluated: sample size calculation, ethical approval, conflicts of interest, blinding, and randomization. Secondly, risk-of-bias assessment was conducted based on ten entries, which aimed to evaluate selection bias, performance bias, detection bias, attrition bias, reporting bias, and other biases [69]. When the contralateral eye was used as the control, questions for random housing and allocation concealment were answered with not-applicable (N/A).

### 2.5. Meta-Analyses

Studies describing non-viral gene therapy did not report sufficient analogous outcome parameters to perform meta-analyses, and were therefore analyzed qualitatively. For studies reporting gene augmentation therapy outcomes, parameters eligible for meta-analysis were a- and b-wave amplitude and b/a-ratio on ERG. Standardized mean differences (SMD; bias corrected Hedges’ *g*) were reported for continuous data. A random-effects model was used to assign weights and compute the overall effect size. To assess between-study heterogeneity, a Chi-square test of homogeneity was performed and I^2^ was reported. Subgroup analyses were performed to assess the effect of input parameters on the overall effect size. The following subgroups contained three or more groups, and were eligible for analyses: mouse model, delivery method, dose, age at injection, and follow-up time point. Publication bias was assessed through analysis of corresponding funnel plots and Egger’s regression-based test. *p*-values below 0.05 were regarded as statistically significant. Statistical analyses were performed using SPSS (IBM, version 28.0.1.1) and graphing was performed using GraphPad Prism 8.

## 3. Results

After the initial literature search, a total of 3982 results were identified from MEDLINE, EMBASE and Web of Science, after which deduplication yielded 2945 results. The inclusion and exclusion procedures are outlined in Figure 2. After title and abstract screening, 76 results were included, and these were split into 41 in vitro and 35 in vivo studies. Subsequent full-text screening yielded 25 total results, giving rise to a new classification split between (1) 19 studies using viral vectors in in vivo models, and (2) 6 studies using alternative and non-viral methods to target *RS1* in vitro as well as in vivo. In the time between the first search and the second, two new studies published after the initial search data were found through other sources and included.

The six studies involving alternative viral and non-viral methods did not have enough analogous outcome parameters to qualify for meta-analysis, and were therefore assessed qualitatively. The 19 studies involving viral vectors reported one of the following results and were further examined in qualitative and/or quantitative analyses: SLO, OCT, ERG, vision-based behavior, and/or IHC. Of these, meta-analyses were performed for ERG b-wave amplitude (*n* = 7, with *n* = 37 experimental groups), a-wave amplitude (*n* = 4 studies describing *n =* 13 experimental groups), and b/a-ratio (*n* = 3, describing *n =* 10 experimental groups). For SLO, OCT, IHC and vision-related behavioral tasks, each outcome parameter was quantitatively reported in less than three studies, and was therefore ineligible for meta-analysis and were therefore described qualitatively.

### 3.1. Adeno-Associated Viral Vector-Mediated Gene Replacement Therapy in Rodent Models for X-Linked Juvenile Retinoschisis

#### 3.1.1. Use of Rodent Models for X-Linked Juvenile Retinoschisis

To date, several mouse and rat models for XLRS have been developed, showing variable disease onset, severity and progression [70,71,72,73,74,75,76]. Both rats and mice have a gene orthologous to human *RS1*, which is referred to as retinoschisis-1 homolog (*Rs1h*). Although no spontaneous mutants have been described, several models have been developed using various methods, including an N-ethyl-N-nitroso-ureum (ENU)-based mutagenesis screen and a targeted (CRISPR-Cas9) *Rs1h* knockout (KO) [70,71,72,73,74,75,76].

Four different Rs1h-deficient rodent models were used in the included studies. Their characteristics are summarized in Table 1. Overall, the pathological characteristics are highly similar within species, but large interspecies differences exist, particularly in disease severity and progression.

#### 3.1.2. Risk of Bias

All included in vivo studies were assessed on their possible risk of bias using the SYRCLE tool [69]. The questions and results of this assessment are summarized in Figure 3. First, the reporting of key quality indicators (ethical approval, conflicts of interest, sample size calculation, blinding and randomization) was assessed (Figure 3A). Secondly, we scored the studies on their risk of selection, performance, detection, attrition and reporting bias using a set of ten questions (Figure 3B). When the contralateral eye was used as control, questions for random housing and allocation concealment were answered with N/A.

Most studies reported on conflicts of interest (83%), and ethical approval and adherence to guidelines for the use of animals in research (78%). In 94%, there was no information on blinding or randomization, and no studies reported on the method of sample size determination (Figure 3A). Blinding and randomization were mentioned in only a small subset of studies (6% each), but details on methodology were not reported. Due to this lack of reporting, most studies were found to have an unclear risk of bias.

#### 3.1.3. Main In Vivo Study Characteristics

For in-depth analysis, we first selected the 19 studies describing AAV-mediated *RS1* gene augmentation therapy in rodents. Main characteristics and reported outcome parameters of the in vivo studies are listed in Table 2.

##### Animal Characteristics

The majority of studies were performed on Rs1h-deficient mice (89%). Two studies used rats (11%). The most commonly used model (63%) was the *Rs1h*-KO mouse developed by Zeng et al. [71] followed by the Weber *Rs1h*^−/y^ mouse [70] (26%) (Table 2). The rat models with a deletion in exon 1 [76] and a deletion in exon 3 [75] were each used in one study. Males were the predominantly used sex: only one study used both sexes [80], and none used only female mice.

In 15 out of 21 experimental groups (71%), animals were treated before they were one month old (median: three weeks (w)) (Figure 4E), while in the remaining studies the animals were injected at older ages (4 w–7 months (m)), when pathology had already developed. Thus, the most commonly used animal characteristics were young male *Rs1h*-KO mice.

##### Therapy Characteristics

All studies reported the viral capsid, serotype, promoter and transgene used (Figure 4A). Some studies used multiple vectors, resulting in 23 experimental groups suitable for our analysis. The AAV8 capsid was used in 10 of 23 (43%) experimental groups. AAV2 and AAV5 were used in 17% and 13%, respectively. Four recombinant capsids derived from AAV2 were each used once. These were, firstly, AAV2tYF, which has three tyrosine residues mutated to phenylalanine (Y to F) residues on the surface of the AAV2 capsid [89]. Secondly, there was AAV7m8, which has a ten-amino acid peptide (7m8) inserted in the AAV2 protein sequence and transfects the inner and outer retina [90]. Thirdly, there was AAV2/4, a recombinant subtype of AAV2 that particularly transduces photoreceptors [91]. Fourthly, rAAV2.2-Y444F was used, which is a rAAV2 serotype 2 with a single mutation (Y444F) on the capsid [34]. One group used AAVshH10, which is derived from the AAV6 shuffled library and primarily targets Müller cells [92].

In 8 out of 23 experimental groups (35%), a ubiquitous promoter was used, which was either chicken beta-actin (CB), cytomegalovirus (CMV), or CAG. CAG consists of the CMV early enhancer, a rabbit beta-globin splice adapter, and the promoter, first intron and exon of CB. Nineteen percent opted for a photoreceptor-specific promoter, either mouse opsin (mOP), or rhodopsin (rho). Thirty-five percent used a modified RS1 promoter coupled with an interphotoreceptor retinoid-binding protein enhancer (scRS/IRBP). One experimental group was treated with a complex termed hRSp4, which consists of the *RS1* promoter, human *RS1* cDNA with a truncated first intron, human beta-globin polyadenylation site, and AAV2 inverted terminal repeats [82]. Finally, one study had two experimental groups using, respectively, a shortened version of bipolar-cell-specific promoters mGluR6 (mini-mGluR6), and a human-DNA MiniPromoter specific to ON bipolar cells, Ple155 [34]. Out of the 20 experimental groups, 15% used mouse *Rs1h* cDNA, whereas 85% used human *RS1* cDNA.

The gene supplementation was delivered by intravitreal injection (IVI) in 14 out of 21 experimental groups (67%) (Figure 4B). The rest used subretinal injection (SRI). The injection volume, whether subretinal or intravitreal, ranged between 1 and 2 µL (mode: 1 µL) (Figure 4C). For IVI, a 35 gauge needle was used most often, in 7 out of the 13 experimental groups that reported needle size (54%) (Appendix A). For SRI, the most popular choice was the 33 gauge (three out of seven experimental groups; 42%). The total amount of virus ranged from 1 × 10^6^ to 7.5 × 10^10^ vector genomes per eye (vg/eye) (median = 4 × 10^9^, mode = 2 × 10^10^ vg/eye) (Figure 4D). The intervention age was variable (5 days (d)–7 m postnatal), but most included studies chose timepoints before the animals were one month old (median = 3 w) (Figure 4E).

Most studies either used an animal’s fellow eye as the control (internal control) or other control animals (external control). One study had two experimental groups: one externally controlled and one internally [82]. Of the 20 experimental groups, 85% had an internal control. Of these internal controls, two were sham-injected [17,71,84], and the rest were directly compared to the untreated fellow eye. In the three groups that were externally controlled, the control animals received a sham injection in one [55] or both eyes [17]. The other group was compared to an untreated animal, in which both eyes received no treatment [82]. Generally, control injections consisted of vehicle [55,84], phosphate-buffered saline (PBS) [71], or a null-vector containing no cDNA [17].

Taken together, the preferred injection method was intravitreal, the most used volume was 1 µL, the most common age at injection was two weeks, the most used titer was 4 × 10^9^ vg/eye or higher and the most common control group was an internal untreated control.

#### 3.1.4. Outcome Parameters and Timepoints

All studies reported at least one of the following outcome parameters following AAV-mediated gene replacement therapy: ERG (*n* = 14; 74%), OCT (*n* = 8; 42%), IHC (*n* = 13; 68%), SLO (*n* = 1; 5%) and/or vision-related behavioral data (*n* = 1; 5%). Both SLO and behavioral data were only reported in one study and were therefore not eligible for meta-analysis but analyzed qualitatively. ERG, OCT and IHC outcomes were reported in three or more studies, and were therefore eligible for meta-analysis. These outcome parameters were reported at variable follow-up timepoints, between two weeks and fifteen months post-injection (PI) (Figure 5). Timing mode and range per outcome parameters were for OCT 12 weeks (2 w–15 m), for ERG 2 months (4 w–15 m), and for IHC 12 weeks (1 w–15 m).

#### 3.1.5. Vision-Related Behavior

One paper described the outcome of vision-related behavioral tasks after AAV-mediated gene augmentation therapy [88]. In this study, mice performed a visually guided swim assay four months post-treatment. Mice were placed in a water maze and scored on their ability to locate a platform. In the dark, Rs1h-deficient mice that were treated subretinally with AAV2/4-CMV-RS1 were able to find the platform significantly faster than untreated controls. Intravitreally treated mice did not perform significantly better compared to controls. Under light conditions, no differences were found between wildtype (WT), untreated and treated animals [88].

#### 3.1.6. Morphological Read-Outs

##### Optical Coherence Tomography

Of the included studies, eight (42%) used OCT to assess treatment efficacy [17,34,54,75,76,85,86,88]. One paper reported SLO data following AAV5-mOP-hRS1 treatment, and showed structural improvements in treated eyes six months PI [77].

Based on OCT data, studies reported the effect of AAV-mediated gene augmentation therapy on cavity formation (100%), laminar integrity (89%) and retinal thickness (50%). Most studies did not quantify these parameters, and showed only the representative images of a single animal, which precludes quantitative comparisons. Nevertheless, most studies reported a reduction in the number and size of cavities, increased or preserved retinal organization and lamination, and overall retinal thickness after treatment with AAV-mediated gene augmentation. These findings are summarized per experimental group in Table 3. The timing of OCT outcome measurements ranged from 2 weeks PI to 15 months PI (median = 14 w). Overall, AAV-mediated gene augmentation therapy decreased the size and number of schisis cavities, improved retinal organization and lamination, and increased the thickness of the retina.

##### Immunohistochemistry

Of the included studies, 13 (68%) reported IHC findings following AAV-mediated gene augmentation therapy, of which 100% confirmed successful transgene delivery by immunostaining for RS1 [17,34,54,76,77,78,79,80,81,82,83,85,88]. The other outcomes assessed were variable and are listed, along with median outcome timepoints, in Table 4. Other outcomes assessed by IHC were the laminar integrity, presence of cavities, microglial or Müller cell alteration, and synaptic pathology. These were all reported in one or two studies each, and can therefore not be meta-analyzed.

#### 3.1.7. Electroretinography

Of the included studies, 14 out of 19 (74%) reported ERG outcomes following AAV-mediated gene augmentation [34,54,55,71,77,78,79,80,81,82,83,85,87,88]. Out of these 13 studies, 4 showed only the ERG waveform of representative animals without mean a- or b-wave amplitude quantification or b/a-ratio [71,79,81,83]. Another study reported ERG data for WT rabbits without Rs1h-deficiency, but not for affected *Rs1h*-KO mice, and therefore did not show a treatment effect [55]. One study was excluded from the meta-analysis as standard deviation could not be derived from the reported data [34]. The remaining eight studies were included for meta-analysis. Experimental details on all groups included in the meta-analyses are outlined in Appendix A.

##### A-Wave Amplitude

In the meta-analysis for a-wave amplitude, four studies describing 13 experimental groups were included [77,85,87,88]. (Figure 6). The input parameters per experimental group are outlined in Appendix A. Group sizes ranged from 4 to 29 (median *n* = 24) eyes for treatment groups, and 12 to 38 (median *n* = 29 eyes) for control groups. Controls were the untreated fellow eye in all but two experimental groups, which used separate untreated animals as controls [87]. SMDs and the 95% confidence interval (CI) were computed per experimental group, and the overall effect size was analyzed with a random-effects model. A Chi-square test for homogeneity found low between-study heterogeneity (*p* = 0.19, I^2^ = 17.7%). Overall, AAV-mediated gene augmentation therapy resulted in a statistically significant moderate increase in the a-wave amplitude of Rs1h-deficient mice (0.75 [0.56, 0.94]). A high or low titer did not significantly impact the overall subgroup effect size (0.79 vs. 0.67, *p*-value = 0.56). Subgroup analyses for the mouse model, delivery method, age at injection and follow-up time point were not performed, as these subgroups would contain less than three groups.

##### B-Wave Amplitude

Out of the 13 studies that reported ERG outcome data, 7 studies describing 37 experimental groups reported averaged b-wave amplitudes of treated mice and controls [54,77,78,80,85,87,88]. The different parameters for each experimental group are listed in Appendix A. Group sizes ranged from 5 to 29 eyes (median *n* = 5) for treatment groups, and 5 to 29 eyes (median *n* = 8) for control groups. Controls were the untreated fellow eye in all except two experimental groups, which used separate untreated animals as controls [87]. The included studies were highly heterogeneous; a test of homogeneity (Chi-square, *p*-value < 0.001) yielded an I^2^ statistic of 76.5%. Due to this high heterogeneity, this meta-analysis and the following subgroup analyses serve for orientational purposes only. Overall, AAV-mediated gene augmentation had a significantly positive effect on b-wave amplitudes (1.44 [1.11, 1.77], *p*-value < 0.001) (Figure 7).

##### Subgroup Analysis b-Wave Amplitude

To assess the potential association of the overall b-wave amplitude effect size with different experimental conditions, effect sizes of several subgroups were compared. We assessed the effect of mouse model (*Rs1h*-KO mouse or *Rs1h*^−/y^ mouse), delivery method (IVI or SRI), low (total dose less than 4 × 10^9^ vector genomes per eye) or high (total dose higher than 4 × 10^9^ vector genomes per eye) titers, age at injection (before or after three weeks of age), and follow-up time point (before or after two months PI) (Figure 8). Other study parameters resulted in subgroups smaller than three groups, and were therefore not performed.

Between the two aforementioned mouse models, *Rs1h*^−/y^ experimental groups achieved higher treatment effect sizes following AAV-mediated gene augmentation, compared to *Rs1h*-KO (1.80 vs. 1.01). Moreover, timing of follow-up measurements showed the highest effect size after two months PI compared to earlier (1.73 vs. 0.97). Injecting before 3 w postnatal slightly increased the effect size (1.70 vs. 1.24). Titers higher than 4 × 10^9^ were found to increase the overall subgroup effect size (1.73 vs. 0.88), and the correlation between b-wave amplitude effect size and dose was significant on meta-regression (*p*-value < 0.05). Meta-regression found no statistically significant correlation between the b-wave amplitude effect size and model (*p*-value = 0.06), age at injection (*p*-value = 0.36), follow-up time point (*p*-value = 0.1) or delivery method (*p*-value = 0.25).

##### B/a-Wave Amplitude Ratio

For b/a-ratio, three studies describing ten experiments were included [82,85,87] (Figure 9). Input parameters per experimental group are outlined in Appendix A. Group sizes ranged from 5 to 29 (median *n* = 25 eyes) for treatment groups, and 5 to 38 (median *n* = 34 eyes) for control groups. Between-study heterogeneity was very high (Chi-square, I^2^ = 90.7%).

Overall, AAV-mediated gene augmentation therapy significantly improved the b/a-ratio in Rs1h-deficient mouse models (1.08 [0.42, 1.74], *p*-value < 0.001). To assess the effect of vector dose (low; less than 4 × 10^9^ vg/eye, or high; more than 4 × 10^9^ vg/eye) on treatment efficacy, a subgroup analysis and meta-regression was performed. A high titer increased the overall subgroup effect size, but meta-regression did not find a statistically significant correlation (1.49 vs. 0.87, *p*-value = 0.09). Subgroup analyses for delivery method, age group and follow-up time point were not performed, as these subgroups would contain less than three experimental groups.

##### Publication Bias

To assess for publication bias in the reporting of the a-wave and b-wave amplitudes, as well as the b/a ratio, we evaluated the funnel plots and performed Egger’s regression-based test. An indication for publication bias was found in the b-wave amplitudes (*p*-value < 0.001), and trim-and-fill analysis predicted missing studies (Figure 10A). No bias was found for b/a-ratio (*p*-value = 0.2), and trim-and-fill analysis predicted no missing studies (Figure 10B). For a-wave amplitudes, trim-and-fill predicted two missing studies, but Egger’s regression-based test found no indication for publication bias (*p*-value = 0.1) (Figure 10C).

### 3.2. Alternative Delivery and Non-Viral Delivery Methods to Target Retinoschisin in Cell-Based and Rodent Models

#### 3.2.1. Use of Alternative Delivery and Non-Viral Methods for X-Linked Juvenile Retinoschisis

Our literature search showed that one out of six studies used an exosome-associated AAV (exo-AAV) as an alternative delivery method to conventional AAVs to target *RS1* [93]. Exosomes offer the possibility of improving AAV neutralization, and therefore transduction [94]. Five out of six studies (83%) used non-viral delivery methods to target *RS1* [58,63,95,96,97]. Of those, one used a non-viral physical method of electroporation [58]. The other four used non-viral chemical methods. Of these four, one used carboxylated nanodiamonds (cNDs) as an inorganic method [97], and the remaining three used organic methods, including solid lipid nanoparticles (SLNs) [63,95] and dual supramolecular nanoparticles (SMNP) [96] (Figure 11). Dual SMNP had a vector size of 110–127 nm [96]. The first set of SLNs had a size of 200–270 nm [95]. This was reduced by Apaolaza et al., also using SLNs, with a particle size between 134 and 167 nm [63]. Exo-AAVs had a particle size of 120 nm [93]. The smallest particle sizes found in our literature search were the cNDs, with a particle size of 3 nm [97] (Figure 11).

In five out of the six included studies, in vitro optimization work was followed by in vivo evaluation. We found a high degree of variability between these studies, due to differences in the cell lines, animal models, or correction and supplementation strategy. Bulk quantitative analysis was not possible, and therefore a qualitative assessment was performed for each separately. An overview of study characteristics is shown in Table 5.

These alternative delivery and non-viral delivery methods have been used to perform either *RS1* integration or *RS1* editing. Three studies used *RS1* integration strategies [63,93,95], and the three other studies targeted *RS1* via CRISPR/Cas9 gene editing [58,96,97].

Two out of the six studies, using an *RS1*-deficient model, studied the rescue of *RS1* in vivo via gene augmentation [63] and in vitro via gene editing [58] using a non-viral method. The remaining four studies evaluated the successful rescue of *RS1* in non *RS1*-deficient models [93,95,96,97]. In particular, they evaluated how alternative delivery and non-viral delivery methods enhance *RS1* transduction efficiency in the retina, providing a proof of concept of these delivery methods to target *RS1*.

#### 3.2.2. In-Depth Analysis of Exo-Adeno-Associated Viral Vector Delivery and Non-Viral Delivery Methods for Retinoschisin Integration

The vectors used for *RS1* integration were exo-AAV [93] and SLNs [63,95]. One type of SLN was used in an in vivo XLRS model to rescue *RS1* via gene augmentation [63].

##### Exo-AAV Delivery Method

The efficiency of exo-AAV *RS1* vectors were investigated in one study as an alternative tool to standard viral delivery systems [93]. Exosomes, lipid bilayer vesicles which mediate communication between cells, offer the possibility of encapsulating AAVs, thereby preventing their degradation and neutralization [98]. Human *RS1* cDNA was introduced into the pAAV2-ZsGreen (*Zoanthus* sp. Green) plasmid, and subsequently encapsulated into the AAV2 vector. This final construct was resuspended in combination with the exosomes into sterile PBS. The constructs AAV2-RS1-ZsGreen and exo-AAV2-RS1-Green were transduced in HEK-293T, ARPE-19 and fibroblast cells (evaluations are summarized in Table 5). After in vitro optimization, transduction efficiency of the exo-AAV2-RS1-ZsGreen vector was evaluated in vivo in C57BL/6 mice (evaluations are summarized in Table 6).

##### Non-Viral Chemical Methods

Successful *RS1* integration in vitro and in vivo was also achieved by using SLNs [63,95]. These lipid-based nanoparticles offer physical stability, protection from degradation, easy composition modification and straightforward production [61,99].

SLNs were tested for the first time as a method to target *RS1* when the ratios of Dextran/Protamine/DNA/SLN were evaluated in vitro, to determine the best transduction efficiency in ARPE-19 cells [95]. The maximum transfection capacity was achieved at 72 h after transfection with the formulation dextran-protamine-DNA-SLN (1:2:1:5). The presence of dextran and protamine in the nanoparticles improved *RS1* transduction and increased *RS1* expression. SNL transduction efficiency carrying exclusively pCMS_EGFP (pCMS = immediate early promoter of CMV, EGFP = enhanced green fluorescent protein) was further evaluated in vivo using Wistar rats. When administered intravitreally, an increase in *EGFP* expression in retinal ganglion cells was observed. After subretinal administration, high *EGFP* expression was detected in the RPE layer [95].

Characterization and rescue of *RS1* using SLNs was first achieved in vivo in 2015 [63]. Initial optimization, involving the replacement of dextran in the formulation by hyaluronic acid, was conducted in vitro. In particular, ARPE-19 cells transduced with the pCEP4 vector, encompassing the CMV promoter and a hygromycin resistance marker, pCEP4-RS1, using the protamine–hyaluronic acid-DNA-SLNinduced RS1 production up to 0.7 ng/mL. This SLNs formulation achieved better results than the dextran-protamine-DNA-SLN formulation previously tested, which induced RS1 production up to 0.2 ng/mL [63,95]. These vectors were dissolved in HEPES buffered saline buffer before transfection. Subsequently, they evaluated the SLN transfection efficiency using the *Rs1h^−/y^* mouse model. Restoration of RS1 resulted in a reduction in PR loss, smaller retinoschisis cavities and improved retinal organization. The experimental settings for the in vitro and in vivo evaluation are summarized in Table 5 and Table 6, respectively.

#### 3.2.3. In Depth Analysis of Non-Viral Delivery Methods to Perform Gene Editing

Electroporation, dual SMNP and cNDs were used to deliver CRISPR/Cas9 constructs to edit *RS1* [58,96,97]. Dual SMNP and cNDs were used in vitro and in vivo as a proof-of-concept to target and edit *RS1*, while electroporation was used in an in vitro XLRS model to edit *RS1*, by correcting or introducing mutations.

##### Non-Viral Physical Method

CRISPR/Cas9 gene editing was used to correct the *RS1* c.625C > T mutation in patient-derived human-induced pluripotent stem cells (hiPSC) [58]. In this study, CRISPR/Cas9 constructs were introduced via electroporation; details are provided in Table 5. Using the same CRISPR/Cas9 editing tools, the mutation was introduced in healthy control cells as a way to validate that *RS1* mutations were indeed responsible for the retinoschisis phenotype. In addition, they corrected the mutation via base editing with editor ABE7.10. Base editing offers the possibility to edit the genome without generating double-stranded DNA breaks, and thereby reduce the risk of producing unwanted insertions or deletions (indels) [100]. The resultant corrected *RS1* mutant hiPSCs were later differentiated into retinal organoids and compared to the non-corrected counterparts. Corrected retinal organoids did not develop the retinoschisis phenotype, and re-established proper RS1 secretion. Further details are presented in Table 5.

##### Non-Viral Chemical Methods

Carboxylated nanodiamonds are carbon-based nanomaterials that have emerged as a promising non-viral strategy due to intrinsic advantages such as bio-compatibility and non-toxicity [101]. Yang et al. managed to fuse cNDs to DNA using mCherry as a connector, and these particles were resuspended in PBS, and later on in bovine serum albumin to avoid particle reaggregation prior to transfection [97]. cNDs were used as a CRISPR/Cas9 delivery method to introduce the c.625C > T mutation in hiPSCs (details in Table 5) and WT mice (details in Table 6), generating a new XLRS model [97]. The resulting mouse retinas reveal characteristic XLRS features, such as abnormalities in the PR layer, and PR structure, as well as an outer retinal layer thickness reduction.

A different approach to introduce CRISPR-Cas9 components was the use of dual SMNPs. SMNPs were used to perform a CRISPR/Cas9-mediated knock-in of *RS1*, presenting a novel therapeutic solution for XLRS [96]. This method is based on the co-delivery of two SMNP vectors, one with Cas9 and single guide RNA (sgRNA) plasmid and one with the donor-*RS1* and green fluorescent protein (GFP)-plasmid. These plasmids were encapsulated by SMNPs containing carbon dot-polyethylenimine (CD-PEI) (to precipitate DNA), adamantane-grafted (Ad) Polyamidoamine (Ad-PAMAN) (to self-assemble the particles), Ad-polyethylene glycol (PEG) (to improve solubility) and Ad-PEG-adenosine-5’-rp-alpha-thio-triphosphate (TAT) (to penetrate the membrane) [96]. SMNPs were resuspended in PBS before transfection [96]. Multiple formulations were optimized in synchronized B16 mouse melanoma cell lines, resulting in successful integration of the 3.0 kb *RS1/GFP* gene 48 h after SMNP treatment (Table 5). SMNPs were also tested in vivo, using albino BALB/c mice via IV injections (Table 6). GFP signals were present from day 18 until day 30, demonstrating the integration of *RS1* in mouse retinas.

#### 3.2.4. In Vitro Readouts

The six included studies performed initial in vitro optimization before in vivo studies were pursued, and the *RS1* transfection capacity was evaluated in vitro. The outcomes used to evaluate the transfection capacity were variable and are included in Table 7 (for studies following the *RS1* integration strategy) and in Table 8 (for studies following the *RS1* gene editing strategy), with the evaluation time points after transfection.

#### 3.2.5. In Vivo Readouts

After in vitro optimization, four out of the six included studies tested these novel delivery methods in vivo. Nevertheless, only one study used an Rs1h-deficient model to rescue *RS1* [63], while the remaining three studies used WT mice in order to evaluate the success of the strategy and delivery method. The outcomes used to evaluate transfection capacity are included in Table 9 (for studies following the *RS1* integration strategy) and in Table 10 (for studies following the *RS1* editing strategy).

## 4. Discussion

In this systematic review, 19 studies that performed AAV-mediated gene therapy in Rs1h-deficient rodent models were systematically reviewed and (meta-)analyzed. The most used vector, AAV8-scRS/IRPB-hRS, has been patented (US10350306B2) and is currently in a clinical trial (ClinicalTrials.gov: NCT02317887), as is rAAV2tYF-CB-hRS1 (ClinicalTrials.gov: NCT02416622) [68,102,103]. In addition, six studies that tested novel alternative and non-viral delivery methods targeting *RS1* as a proof of concept were qualitatively analyzed. While no clinical trials have been instigated for these methods at this time, exo-AAV2-ZsGreen has been patented (CN111500634B).

### 4.1. The Preferred Practices of In Vivo Study Characteristics

The majority of analyzed studies used young male mice, in line with clinical relevance. We identified one study that used both male and female mice and none that used aged mice (18–24 months). Four different Rs1h-deficient animal models were used. The preferred model was the *Rs1h*-KO mouse [71], closely followed by the *Rs1h*^−/y^ mouse model [70]. Overall, these mice exhibited similar disease phenotypes [104]. The most commonly tested vector, AAV8-scRS/IRPB-hRS, was the only vector evaluated in both rats and mice. However, as both rat models were developed very recently (2022), we may expect more vectors to receive further testing in rat models.

### 4.2. In Vivo Study Quality Assessment Reveals Insufficient Self-Reporting

To assess the quality of analyzed in vivo AAV studies, several tests and analyses were performed. SYRCLE’s tool for assessing quality and risk of bias revealed an unclear risk of bias for most categories due to insufficient reporting of the methodologies applied. As a consequence, this makes it difficult to weigh the findings in these meta-analyses accurately. Moreover, differences in study quality can introduce heterogeneity in our meta-analyses. For the subset of studies included in the meta-analyses for a- and b-wave amplitude, publication bias was present, as shown by trim-and-fill analyses and Egger’s regression tests. In particular, the b-wave amplitude effect size funnel plot was highly asymmetrical. Often, this is due to underreporting of negative or statistically insignificant results, swaying the results of this overall effect size analysis and perhaps indicating a possible reason for the translational limitations.

Differences in control groups may contribute to between-study heterogeneity. Most experimental groups were internally controlled. The majority of control eyes or control groups were untreated. However, the trauma of the (subretinal) injection itself can have trophic effects in the retina [105], and therefore sham control injections are recommended.

### 4.3. In Vivo Qualitative Analysis Shows Variability in Outcome Parameters

In eight of the analyzed studies, the success of AAV-mediated gene augmentation therapy on retinal morphology was evaluated by OCT. Among the studies reporting OCT data, most reported the same parameters (cavity reduction, retinal thickness, lamination integrity). However, of the eight studies describing 18 experimental groups reporting OCT data, the majority reported treatment success without quantification, showing only visual confirmation of smaller cavities in representative images. This precludes direct comparisons of treatment success.

While schisis cavities are the hallmark of XLRS, not all studies tested for cavity formation. In one case, while animals were treated at young ages (P14), they were assessed for treatment success only at 14 months of age, at which time cavities will ordinarily have already disappeared in untreated animals [79]. In other cases, the treatment effect on retinal structure was investigated using IHC. In vivo imaging of retinal integrity and presence of cavities is preferable to drawing conclusions of retinal integrity over post-mortem IHC. After all, OCT imaging is non-invasive and cannot introduce tissue damage or artifacts via the technique itself. The process of tissue sectioning can introduce additional damage to an already-fragile retina, falsely exacerbating the disorganization of the retinal layers. Ideally, the two techniques are both performed and compared, as each can provide unique information. Kjellstrom et al. demonstrated the necessity of confirming transgene delivery on IHC. In seven injected animals, large variation was observed in outcome. Post-mortem IHC staining for RS1 showed that animals with lower or no rescue had lower or absent RS1 protein presence in the retina [79]. Thus, the testing and reporting of technical treatment success by confirming exogenous RS1 expression is essential to correctly interpret (mixed) results.

### 4.4. Effect-Size Analyses Show a Significant Effect of Adeno-Associated Viral Vector-Mediated Gene Therapy on a- and b-Wave Amplitudes, and b/a-Ratio

The meta-analyses confirmed the efficacy of AAV-mediated gene augmentation therapy on the ERG b-wave amplitudes (1.44 [1.11, 1.77]), a-wave amplitudes (0.75 [0.56, 0.94]), and b/a-ratio (1.08 [0.42, 1.74]). However, between-study heterogeneity of the studies included for the b-wave amplitude and b/a-ratio were high (I^2^, respectively equaled 76.5% and 90.7%). This was to be expected, as we included a variety of animal models, gene therapy vectors, intervention ages, and follow-up timepoints. Nevertheless, due to the high heterogeneity, these analyses, and the subgroup analyses that followed, can serve for orientational purposes only.

#### 4.4.1. Higher Titers Increase Therapy Effect on b-Wave Amplitude

To understand the possible effects of the study characteristics on treatment efficacy, several subgroup analyses were performed. In all three meta-analyses, subgroup analysis for dose was performed. Titers higher than 4 × 10^9^ vg/eye had a positive effect on the subgroup effect size in all analyses, but metaregression found that this was statistically significant only for b-wave amplitude.

#### 4.4.2. Considerations When Choosing a Delivery Method

When targeting the outer retina, SRI is most often the delivery method of choice. This is likely why the first few studies attempting AAV-mediated gene supplementation in XLRS models used SRI [71,77,78]. Interestingly, the Rs1h-deficient mouse retina is particularly suited to intravitreal delivery of a vector aimed to act at the outer retina, perhaps due to the disorganized state of the Rs1h-deficient retina, allowing for more easy diffusion across the retina [82]. Moreover, the process of SRI in an already-fragile retina can introduce many complications, such as retinal detachment, bleeding, inflammation, and vector leakage [77,106]. Thus, IVI became the delivery method of choice in animal models for XLRS, as evidenced by 14 out of 19 included studies using IVI. In this meta-analysis, b-wave amplitude was the only parameter where subgroup analysis for delivery method was possible. While IVI injection led to a higher subgroup effect size, meta-regression did not show significant differences. The higher effect size of IVI might also be explained by the nature of the technique, as SRI treats locally, in contrast to IVI, and ERG responses were recorded full-field.

Nonetheless, the failure of IVI delivery in human clinical trials has necessitated the re-evaluation of the delivery method, also because SRI appears to be generally safe and effective in humans [107]. One study directly compared IVI and SRI delivery for vector AAV2/4-CMV-hRS1 in the *Rs1h*-KO mouse, and found that SRI led to superior rescue on all measured parameters [88]. In fact, IVI delivery did not manage to transfect the ONL, whereas SRI did [88]. Some final advantages of SRI over IVI are that SRI induces less immunogenicity than IVI, and is more effective at targeting PRs, the main cell population of interest in XLRS [108].

#### 4.4.3. Considerations When Choosing an Intervention Age

XLRS patients often show symptoms at very young ages and will likely have a significant degree of pathology already present at the time they are seen for treatment [4]. The ideal therapy would prevent the disease from developing, or cure the disease when it is already symptomatic. In many studies, the biggest improvement in cavity formation was achieved by delivering a vector in young pups. Injection at P15 led to significant and progressive improvements in scotopic and photopic retinal function and morphology for AAV5-mOP-hRS1 [77,80], AAV2/2-CMV-Rs1h [79], and AAV8-scRS/IRBP-hRS [17]. Similarly, in rats, the preventative application AAV8-scRS/IRBP-hRS at P7 was shown to reduce cavity formation on OCT [76]. Intervention at early time points has been shown to have long-lasting effects, as a single injection at two weeks old with the AAV(2/2)-CMV-Rs1h vector effectively reduced the disease phenotype up to 14 months [79].

The vectors were also tested at later time points, and applied at one month of age or later, when pathology was well developed. Injection with AAV8-scRS/IRBP-hRS at P30 prevented cavity formation and loss of lamination, but not thinning of the ONL [83]. The delivery of AAV2/4-CMV-RS1 between P60 and P90 led to the disappearance of cavities two weeks later, and functional improvement 50–60 days PI [88]. Two analyzed studies directly tested and compared multiple intervention timepoints. Comparing injection at P14 and P30 showed that the scotopic b-wave amplitude rescue at 4 months PI was lower in animals injected at P30 for all vectors tested (AAV7m8-rho-hRS1 and AAVshH10-CAG-hRS1) [54]. Similarly, AAV5-mOP-hRS1 was used to treat mice at P15, 1 m, 2 m and 7 m, which showed that treatment efficacy declined with the age at intervention [80]. However, injection as late as 2 m was still able to significantly improve retinal function [80]. Thus, while a younger age at intervention seems to have a more pronounced effect, there still appears to be a treatment effect at older ages.

It is important to take into account that the peak expression of AAV-delivered genes is 4–6 weeks post-injection [109]. As apoptotic events in the ONL peak at P18 [110], and exogenous *RS1* gene expression was observed as early as at seven days PI [17] and RS1 protein at two weeks PI [77], injection at P15 may already be too late to prevent cell death in the ONL. Furthermore, it is crucial to consider species differences in disease progression when choosing an intervention time window.

#### 4.4.4. Treatment Effect Size Was Increased When Evaluated at Two Months or Later

Overall, a treatment effect was not immediately perceived on ERG or OCT, even if exogenous protein is present in the retina already at two weeks PI. In two studies, ERG amplitudes were not significantly different between treated and untreated eyes at one month PI when delivered at P15. At two and three months PI, the ERG waveform in treated eyes was markedly improved, with amplitude improvements peaking at three months and sustaining up to five months PI [77,78]. This indicates a delay in the treatment effect on retinal function. Clearly, restoration of RS1 protein production does not immediately result in a functional effect at cell surfaces. In line with these findings, the effect size was positively affected when b-wave amplitudes were assessed two months PI or later. However, improvements in cavity size and number were described as soon as two weeks PI [88], indicating that cavity resolution might be required for functional improvement.

### 4.5. Discrepancies between Structural and Functional Readouts

The XLRS disease phenotype can be generally divided into two categories: structural and functional. The ideal therapy resolves cavity formation, protects the integrity of lamination, prevents ONL degradation and rescues retinal function. We found that an improvement in structure does not always guarantee a rescue in function. Post-developmental adult vector delivery at 13 weeks was able to restore RS1 expression and rescue the ERG waveform and improve the scotopic b-wave by 6 months of age, but it did not lead to any improvements in retinal morphology or photoreceptor survival [71].

### 4.6. Adeno-Associated Viral Vector Immunogenicity Considerations

Some safety and tolerability tests were performed for a subset of the included AAV-mediated gene augmentation therapies. Safety and tolerability were tested for AAV8-hRS/IRBP-hRS1 in *Rs1h*-KO mice and WT rabbits [55,111], and for AAV2tYF-CB-hRS1 in *Rs1h*^−/y^ mice [84]. Both treatments were well tolerated, the vector presence was limited to the injected eye, and no adverse effects, aside from minor increases in immune activation, were observed. For AAV8-hRS/IRBP-hRS1, vitreous inflammation was observed in the treated eye of WT rabbits two weeks after treatment, with a duration proportional to the vector dose, but resolved within three months and did not lead to irreversible tissue damage [55,111]. The AAV8 capsid, but not the *RS1* cDNA that was delivered, was shown to elicit these immune responses in rabbits [55].

Although AAVs are generally regarded as safe, several concerns about AAV vector immunogenicity remain unaddressed [65,108,112]. This is likely why efforts have been made to move away from viral vectors to non-viral in recent years. Non-viral vectors are easy to produce and customize, and are less restricted in cargo capacity [113]. While non-viral vectors have been demonstrated to have lower immunogenicity and toxicity, they generally also show lower transduction efficiency; as a consequence, further optimization is still required [65,114].

### 4.7. Alternative Viral and Non-Viral Strategies as Novel Methods to Integrate or Edit Retinoschisin

The lack of success of viral vectors to alleviate symptoms in clinical trials has been partly attributed to insufficient AAV transduction and, as a consequence, the ability to transcribe [93,114,115,116]. Our literature search revealed several alternative, usually non-viral, delivery methods for *RS1* or CRISPR-Cas9 components targeting *RS1*: SLN, dual SMNP, cNDs, electroporation, and exosome-associated AAVs. The preliminary optimization of non-viral and alternative delivery methods has been conducted in vitro in all included studies, in order to reduce costs, enable direct evaluation of the desired factor, and to have less ethical restrictions.

The studies that introduced *RS1* evaluated its integration within the cells of interest. In the included studies, the cell models used were HEK-293Ts, hiPSCs and ARPE-19. HEK-293T cells and hiPSCs are characterized by a fast division rate, in contrast to slow-dividing ARPE-19 cells [117]. Cell dynamics are key to ensure a successful transduction, which is a process facilitated by cell division [117]. In contrast, cells with slow or no division (such as PRs and RPE cells) develop a nuclear membrane which is harder to cross, hindering transduction [117]. Therefore, the assessment of new non-viral delivery methods in different cell lines with different cell dynamics is key to evaluate the full potential of the desired treatment.

Particle size and charge are relevant parameters for the successful in vivo integration of *RS1* or CRISPR-Cas9 components [58]. ILM thickness, pore size and negative charge directly affect nanocarrier diffusion [46]. Moreover, ILM pore size differs between species and significant changes in ILM morphology have been reported with advancing age in humans [118]. This highlights the need for an in-depth evaluation of different compositions, designs, and particle sizes for nanocarriers in order to extrapolate in vivo results to clinical trials.

In order to evaluate the successful *RS1* integration during in vitro optimization, in vitro transfection capacity has been assessed [63,93,95]. Different techniques were used for this purpose: quantification of % GFP positive cells, GFP detection, and mRNA and protein expression. While the studies aimed to evaluate the same parameter, they used different readouts. As a consequence, a direct comparison is prevented. Moreover, the post-transduction evaluation time points across studies were different. Evaluation time points are crucial to determine the peak of transduction efficiency, and the duration of sustained expression.

Overall, the qualitative evaluation of the non-viral and alternative delivery vectors revealed insufficient reporting of applied methodologies, as well as high variability within the selected outcomes. This echoes the message reported in the in vivo AAV-mediated gene augmentation analysis. If novel non-viral and alternative delivery methods are evaluated along similar parameters as existing options, direct comparisons can be made, and treatment efficacy can be accurately weighed. Therein lies the importance of systematic reviews and meta-analyses like this one, in order to extract the preferred practices and help design new studies, enabling systematic comparisons.

## 5. Conclusions

This systematic review summarized and analyzed the currently available preclinical data on gene therapy targeting *RS1* in cell-based and Rs1h-deficient rodent models. The meta-analyses on a-wave and b-wave amplitudes and b/a-ratio on ERG confirmed the efficacy of AAV-mediated gene augmentation therapy in Rs1h-deficient rodents. Unfortunately, the quality assessments indicated the presence of risks of attrition, detection, performance, selection, and publication biases, emphasizing the need for increased transparency and better reporting in animal research. Between-study heterogeneity, as well as the presence of biases, may result in an overestimation of therapy effectiveness. This, combined with species differences and potentially suboptimal delivery methods, might be a reason why these therapeutic candidates did not yet achieve success in clinical trials. Therefore, we encourage the development of standardized approaches and transparency in experimental setup, to ensure the increased translatability and reproducibility of future gene therapy approaches for XLRS. Moreover, we found promising proof-of-concept work exploring alternative viral and non-viral approaches for gene therapies targeting *RS1*. While these particular avenues of XLRS therapy may still be in their infancy, we expect the quantity of these types of studies to increase significantly in the coming years. As a result, new treatment options may arise, as well as novel insights into XLRS.

Overall, this systematic review of gene therapy in rodent and cell-based models of XLRS highlights the potential for gene therapy in XLRS, and at the same time stresses that care needs to be taken in preclinical study design and reproducibility.

## Figures and Tables

**Figure 1 ijms-25-01267-f001:**
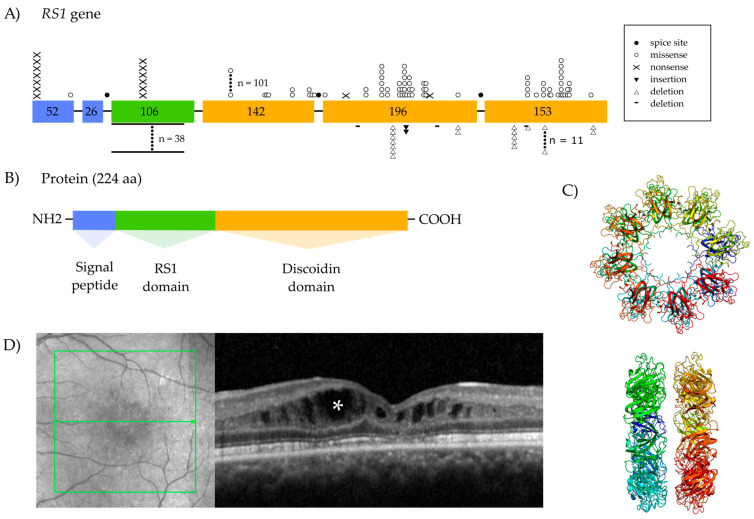
Schematic structure representing the retinoschisin (*RS1*) gene (**A**), protein diagram (**B**), the quaternary structure of the protein (**C**), and hallmarks of X-linked juvenile retinoschisis (XLRS) (**D**). (**A**) (Adapted from Hahn et al. [4]). An overview of *RS1* shows the localization and variants within the gene. Colored boxes represent the six exons, and the numbers within the boxes represent their size in nucleotides. The type of variant is presented in the legend, and the frequency and distribution are described within the gene structure. Splice site (*n* = 38), missense (*n* = 101), and deletion variants (*n* = 11) are the three most common variants. (**B**) Overview of the RS1 protein, featuring the signal peptide domain, the RS1 domain, the discoidin domain and the C-terminal domain. (**C**) The top and side view of the octamer and double octamer. (**D**) Fundus photograph of a 35-year old patient with a hemizygous c.428A > T p.(Asp143Val) mutation in the RS1 gene patient with XLRS, and optical coherence tomography image showing macular schisis (asterisk) in this patient with XLRS.

**Figure 2 ijms-25-01267-f002:**
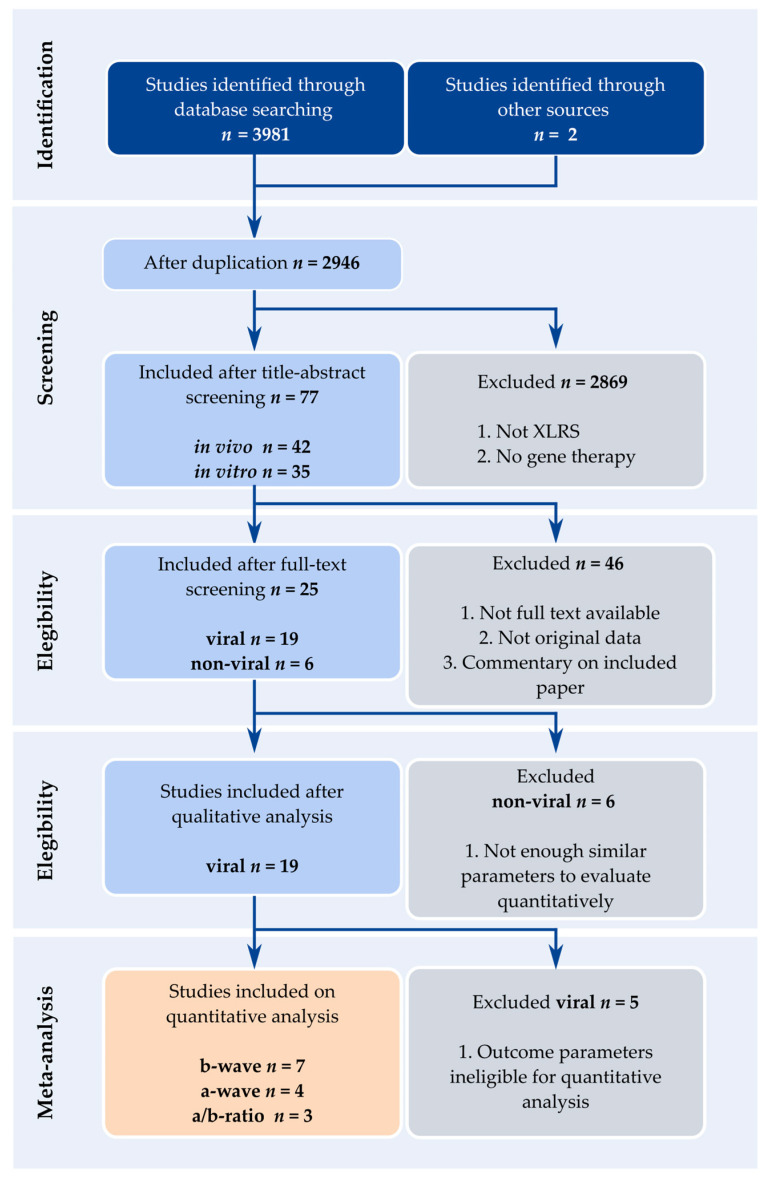
Flowchart outlining the screening and selection process. The selection process was performed through the following stages: identification, screening, eligibility criteria and meta-analysis. Taken together, 3983 studies were identified in the first stage, and after removing duplicates, 2946 studies were included in the screening phase. After title and abstract screening, 77 studies were included. After full-text screening, 25 studies were included, split into 19 studies describing viral methods in in vivo models, and 6 studies describing non-viral and alternative delivery methods in vitro and in vivo. After the second eligibility stage, 19 studies using viral methods were eligible for quantitative analysis. For meta-analysis, seven studies were included for b-wave amplitude, four studies for a-wave amplitude and three studies for the ratio between b- and a-wave amplitudes (b/a-ratio). Four studies were not included in any meta-analysis.

**Figure 3 ijms-25-01267-f003:**
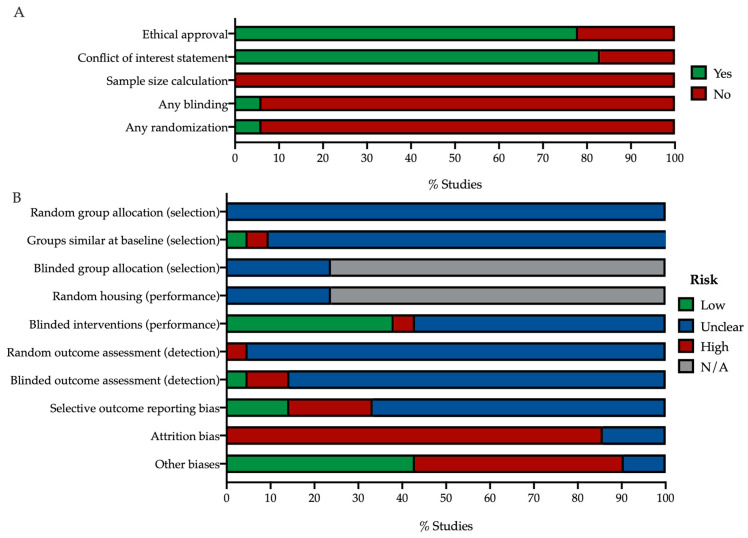
All in vivo studies were assessed on their risk of bias using the SYRCLE tool [69]. (**A**) The studies were assessed on whether any information was provided about ethical approval, conflicts of interest, calculation of sample sizes, blinding, or randomization. (**B**) The studies were assessed on possible risk of selection, performance, detection, outcome reporting, attrition and other biases. When the contralateral eye was used as the (internal) control, questions for random housing and allocation concealment were answered with not-applicable (N/A). Most studies received an unclear risk-of-bias score, due to a lack of reporting of essential quality indicators.

**Figure 4 ijms-25-01267-f004:**
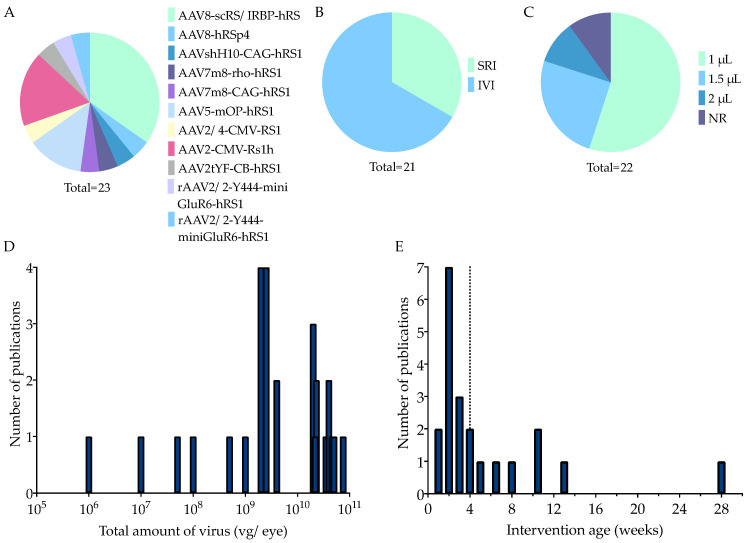
Therapy characteristics of adeno-associated viral vector-mediated (AAV) gene augmentation therapies in included studies. The total number of experimental groups varies per parameter, as some studies tested multiple versions of the same parameter. (**A**) A spectrum of AAV types was used. The total number of experimental groups was 23. (**B**) The majority (67%) delivered the treatment by intravitreal injection. (**C**) Injection volumes ranged from 1 to 2 µL, although 1 µL was most commonly used (55%). (**D**) A number of publications reported different total doses of virus delivered to the eye (viral genomes/eye), but the majority chose a titer >4 × 10^9^ vg/eye. The dotted line indicates the median. (**E**) The animals were treated at variable ages (5 days–7 months), although most were treated before one month of age (median: three weeks; dotted line). scRS/IRBP = modified RS promoter coupled with an interphotoreceptor retinoid-binding protein enhancer; CMV = cytomegalovirus promoter, hRSp4 = a complex consisting of RS1 promoter, human *RS1* cDNA with a truncated first intron, human beta-globin polyadenylation site, and AAV2 inverted terminal repeats; shH10 = AAV subtype derived from AAV6; CAG = promoter element consisting of CMV early enhancer, rabbit beta-globin splice adapter, and the promoter, first intron and exon of CB; CB = chicken beta-actin; RS1 = retinoschisin; *hRS1* = human RS1; rho = rhodopsin; mOP = mouse opsin; Rs1h = rodent homolog to RS1; 2tYF = AAV subtype with three tyrosine residues mutated to phenylalanine on the AAV2 capsid surface; rAAV2/2-Y444F = AAV subtype with a single Y mutated to F on the AAV2 capsid surface; mini-mGlu6 = a shortened version of the ON bipolar cell-specific promoter mGlu6; Ple155 = a human-DNA MiniPromoter specific to ON bipolar cells; SRI = subretinal injection; IVI = intravitreal injection; NR = not reported.

**Figure 5 ijms-25-01267-f005:**
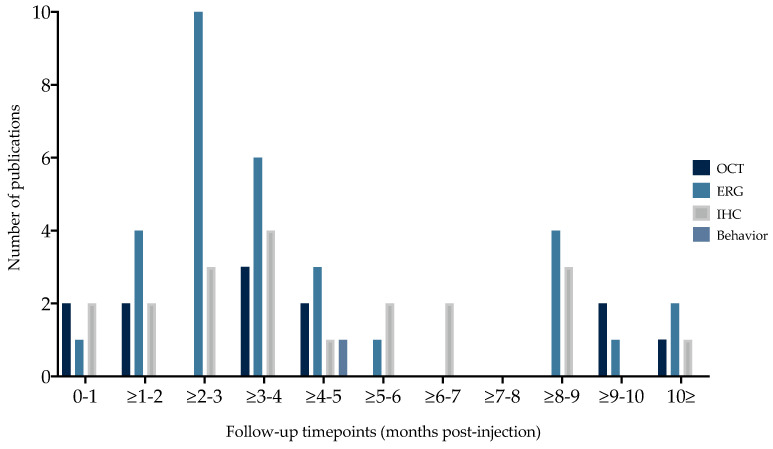
Follow-up time points per outcome parameter selected for in-depth analysis. Outcome parameters were reported at variable timepoints post-injection. Timing mode and range per outcome parameters were, respectively, OCT 12 weeks (2 w–15 m), ERG 2 months (4 w–15 m), and IHC 12 weeks (1 w–15 m). OCT = optical coherence tomography, ERG = electroretinography, IHC = immunohistochemistry.

**Figure 6 ijms-25-01267-f006:**
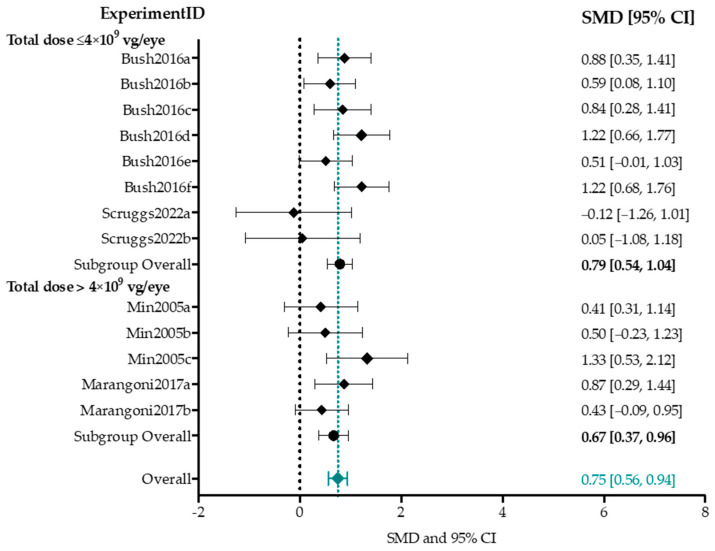
The overall effect size of adeno-associated viral vector (AAV)-mediated gene augmentation therapy on a-wave amplitudes with integrated dosage subgroup analysis. The studies included in this meta-analysis were assigned study IDs according to First Author Publication year, and experimental groups within one publication are distinguished by labeling them a through z. The studies included in this meta-analysis were Bush2016 [85], Scruggs2022 [88], Min2005 [77], and Marangoni2017 [87]. The results are staggered per subgroup (total dose smaller than 4 × 10^9^ viral genomes/eye (vg/eye) and larger than 4 × 10^9^ vg/eye). The black circles represent the overall subgroup effect size, and the green diamond represents the overall weighted effect size. The black dashed line indicates the null effect. The left side of the null effect line favors the control, whereas the right side of the null effect line favors treatment. The overall effect size (0.75 [0.56, 0.94]) was computed with a random effects model and is represented by a green diamond and dashed line. Included studies were assessed for homogeneity with a Chi-square test and were not found to be highly heterogeneous (I^2^ = 17.1%). SMD = standardized mean difference, CI = confidence interval.

**Figure 7 ijms-25-01267-f007:**
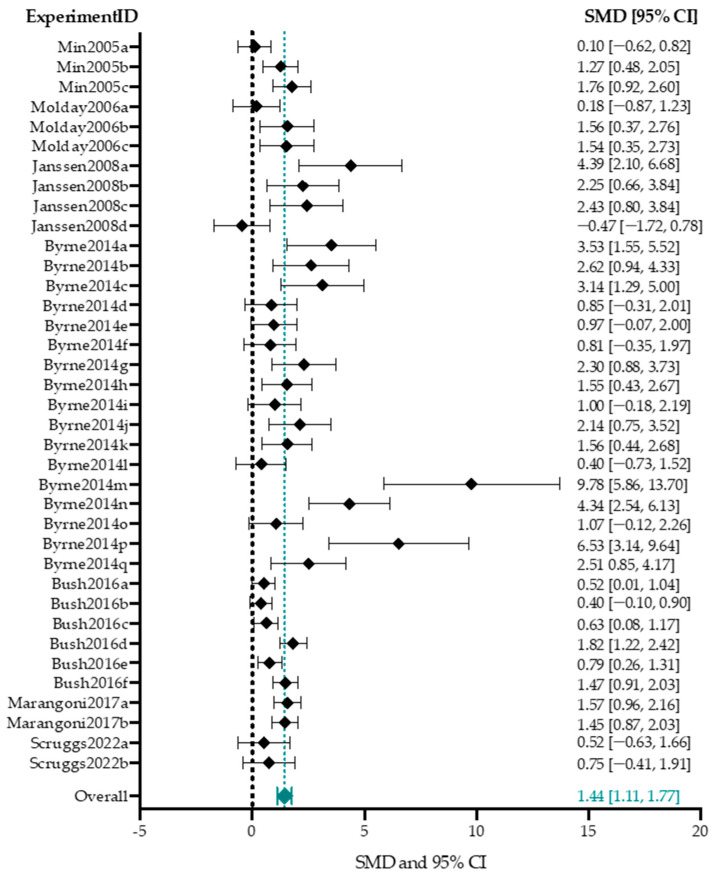
The overall weighted effect size of adeno-associated viral vector (AAV)-mediated gene augmentation therapy on the b-wave amplitude. The studies included in this meta-analysis were assigned study IDs according to First Author Publication year, and experimental groups within one publication are distinguished by labeling them a through z. The studies included in this analysis were Min2005 [77], Molday2006 [78], Janssen2008 [80], Byrne2014 [54], Bush2016 [85], Marangoni2017 [87], and Scruggs2022 [88]. The standardized mean differences (SMDs) with 95% confidence intervals (CI) of each experimental group are displayed in a forest plot. The dashed black line represents no treatment effect. Values left of the null effect line favor control, whereas values right of the null effect line favors treatment. The overall effect size (1.44 [1.11, 1.77]) was computed with a random-effects model and is represented by a green diamond and dashed line. Included studies were assessed for homogeneity with a Chi-square test and were found to be highly heterogeneous (I^2^ = 76.5%).

**Figure 8 ijms-25-01267-f008:**
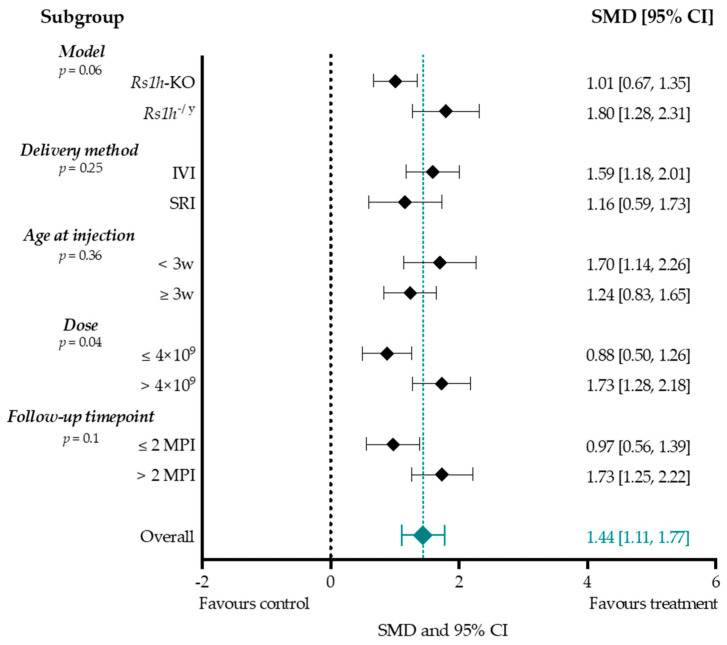
The effect sizes of adeno-associated viral vector (AAV)-mediated gene augmentation therapy on the b-wave amplitude for subgroups mouse model, delivery method, dose, age at injection, and follow-up time point. The black dashed line indicates the null effect. Values left of the null effect line favor the control, whereas values right of the null effect line favor the treatment. The overall effect size (1.44 [1.11, 1.77]) was computed with a random-effects model and is represented by a green diamond and dashed line. The between-study heterogeneity was high (I^2^ = 76.5%), and therefore these subgroup analyses serve for orientational purposes only. *Rs1h* = retinoschisin, KO = knock-out, IVI = intravitreal injection, SRI = subretinal injection, M = months of age, MPI = months post-injection, SMD = standardized mean difference, CI = confidence interval.

**Figure 9 ijms-25-01267-f009:**
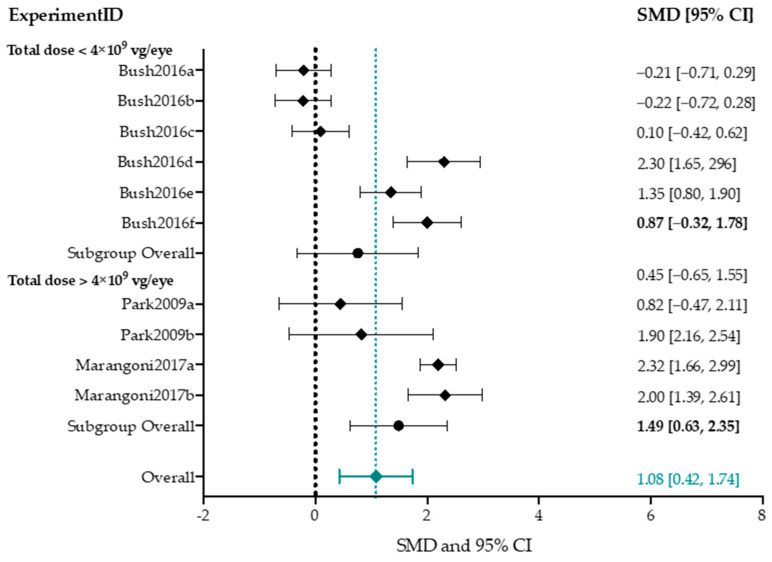
The overall weighted effect size of AAV-mediated gene augmentation therapy on the ratio between the b- and a-wave amplitudes (b/a-ratio). The studies included in this meta-analysis were assigned study IDs according to First Author Publication year, and experimental groups within one publication are distinguished by labeling them a through z. The studies included in this analysis were Bush2016 [85], Park2009 [82], and Marangoni2017 [87]. The results are staggered per subgroup (total dose smaller than 4 × 10^9^ viral genomes/eye (vg/eye) and larger than 4 × 10^9^ vg/eye). The black circles represent the overall subgroup effect size, and the green diamond represents the overall weighted effect size. The black dashed line indicates the null effect. Values left of the null effect line favor control, whereas values right of the null effect line favors treatment. The overall effect size (1.08 [0.42, 1.74]) was computed with a random-effects model and is represented by a green diamond and dashed line. The black circles indicate the overall subgroup effect size. A Chi-square test of homogeneity revealed high levels of between-study heterogeneity (I^2^ = 90.7%). SMD = standardized mean difference, CI = confidence interval.

**Figure 10 ijms-25-01267-f010:**
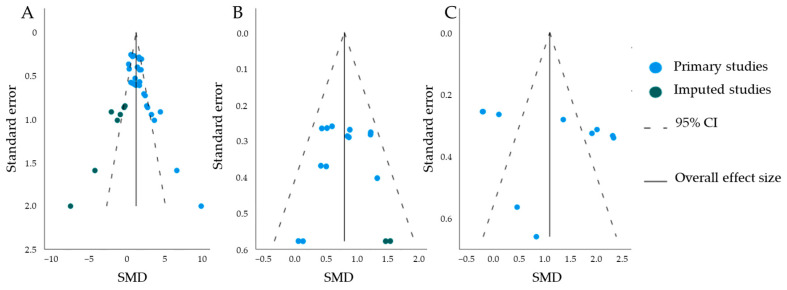
The funnel plots of the meta-analyzed studies. The blue dots indicate standardized mean differences (SMD) of primary studies; the green dots indicate missing studies. The black lines indicate overall effect size, and the dashed lines indicate the 95% confidence intervals (CI). (**A**) The funnel plot for b-wave amplitude. A trim-and-fill analysis predicted missing studies. (**B**) The funnel plot for a-wave amplitude. A trim-and-fill analysis predicted missing studies. (**C**) The funnel plot for the ratio between the b- and a-wave amplitudes (b/a-ratio). Trim-and-fill analysis predicted no missing studies.

**Figure 11 ijms-25-01267-f011:**
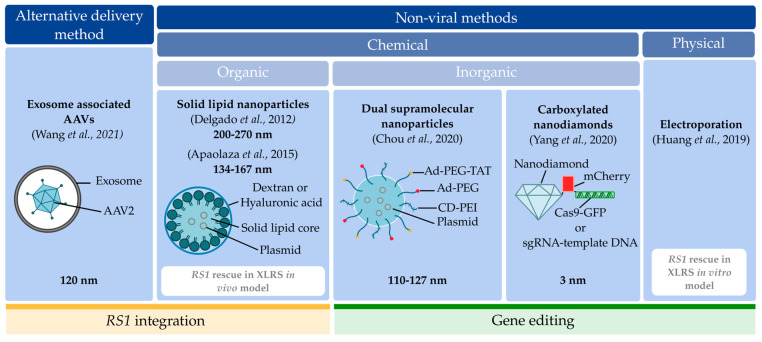
The classification of alternative delivery and non-viral delivery methods. The first classification is alternative delivery and non-viral delivery methods. In the former, the exosome-associated adeno-assiocated viral vectors (AAVs) are classified [93]. Within the non-viral methods category, a second classification can be made: chemical and physical methods. The physical methods contain the electroporation method [58], while the chemical methods can be divided into organic (solid lipid nanoparticles [63,95]) or inorganic (dual supramolecules nanoparticles [96] and carboxylated nanodiamonds [97]) vectors. In each category, a schematic representation of the vectors used for *RS1* or gene editing constructs delivery is included to illustrate the building blocks used in each vector. Dual supramolecular nanoparticle and carboxylated nanodiamond representations have been adapted from Chou et al. and Yang et al., respectively [96,97]. Outlined with a yellow line are the studies designed for gene augmentation, whereas those designed for gene editing are marked with a green line. XLRS = X-linked juvenile retinoschisis, *RS1* = retinoschisin, ad = adamantane-grafted, PEG = polyethylene glycol, TAT = adenosine-5’-rp-alpha-thio-triphosphate, CD-PEI = carbon dot-polyethylenimine, GFP = green fluorescent protein, sgRNA = single guide RNA.

**Table 1 ijms-25-01267-t001:** The four rodent models for XLRS that were used in the included studies. The rodent models were assigned a name based on the names used in the first publication that described the model. In the case of the rat model, a number was added to distinguish the two. XLRS = X-linked juvenile retinoschisis, RS1 = retinoschisin, *Rs1h* = homolog of RS1 in rodents, KO = knock-out, bp = base pair, kb = kilobase, IHC = immunohistochemistry, WB = Western blot, INL = inner nuclear layer, ONL = outer nuclear layer, OPL = outer plexiform layer, PR = photoreceptor, IS = inner segments, OS = outer segments, ERG = electroretinography, P = postnatal day, b/a-ratio = ratio between the b- and a-wave amplitudes.

Name [Ref.]	Species	Strain	Mutation	RS1 Protein Absence	Main Pathological Features	Age of Schisis Onset	Schisis Progression
*Rs1h*^−/y^ [70]	mouse	C57BL/6	Replacement of exons 3 and 4 and intron 3 by a lacZ-neo cassette.	Confirmed on IHC and WB	Schisis cavities in the inner retina are evenly dispersed throughout the central en and peripheral retina. Loss of laminar integrity in the OPL, INL and PR layer. Cell death is restricted to PRs. The a- and b-wave amplitudes are affected, b-wave more so.	P18	Not reported
*Rs1h*-KO [71]	mouse	C57BL/6	Replacement of exon 1 and 1.6 kb of intron 1 by a neomycin resistance cassette	Confirmed on IHC and WB	Schisis cavities are found throughout the retina. Lamination of the OPL, INL and PR layer is affected. The a- and b-wave amplitudes are diminished, and the b-wave is more strongly affected.	P14	Cavities collapse from 4–12 months old
*Rs1h*^−/y^ rat (1) [75]	rat	Long Evans	Deletion of 310 base pairs, encompassing exon 3 and parts of introns 2 and 3.	Confirmed on IHC and WB	Large cavities are observed at P15. Lamination is disrupted, there is cell loss in ONL, and OPL and IS/OS-zone are thinner. The a- and b-wave amplitudes are reduced, as is the b/a-ratio.	P15	Cavities rapidly collapse at P21–P28
*Rs1h*^−/y^ rat (2) [76]	rat	Long Evans	Deletion of 769 bps, encompassing exon 1.	Confirmed on IHC and WB	Large cavities are observed at P15 that are resolved by P30. The ONL is thinner, IS/OS are shorter and OPL is disordered. B- and a-wave amplitudes are decreased.	P15	Cavities have collapsed by P30

**Table 2 ijms-25-01267-t002:** Main characteristics and gene supplementation therapy outcome parameters of the included in vivo studies. The studies were assigned a study ID formatted as First-AuthorPublicationyear. Outcome parameters were only regarded as “yes” (Y) if they were reported in the context of gene supplementation therapy experiments. KO = knock-out; AAV = recombinant adeno-associated viral vector; shH10 = Müller-cell specific serotype derived from the AAV6 shuffled library; 2tYF = AAV2-derived serotype with three tyrosine (Y) residues mutated to phenylalanine (F); hRSp4 = a complex consisting of the RS1 promoter, human RS1 cDNA with a truncated first intron, human beta-globin polyadenylation site, and AAV2 inverted terminal repeats; mOP = mouse opsin; rho = rhodopsin; CB = chicken beta-actin; CMV = cytomegalovirus; CAG = complex consisting of CMV early enhancer (C), chicken beta-actin promoter, first intron and first exon (A), and rabbit beta-globin splice adaptor (G); IRBP = interphotoreceptor retinoid-binding protein; RS1 = retinoschisin; hRS1 = human RS1; Rs1h = RS1 homolog; mini-mGlu6 = a shortened version of the ON bipolar cell-specific promoter mGlu6; Ple155 = a human-DNA MiniPromoter specific to ON bipolar cells; SRI = subretinal injection; IVI = intravitreal injection; SLO = scanning laser ophthalmoscopy; OCT = optical coherence tomography; ERG = electroretinography; IHC = immunohistochemistry; N = no; Y = yes).

Study ID	Species	Model	Treatment	Delivery Method	SLO	OCT	ERG	Behavior	IHC
Zeng2004 [71]	mouse	*Rs1h*-KO	AAV2-CMV-Rs1h	SRI	N	N	Y	N	N
Min2005 [77]	mouse	*Rs1h* ^−/y^	AAV5-mOP-hRS1	SRI	Y	N	Y	N	Y
Molday2006 [78]	mouse	*Rs1h* ^−/y^	AAV5-mOP-hRS1	SRI	N	N	Y	N	Y
Kjellstrom2007 [79]	mouse	*Rs1h*-KO	AAV2-CMV-Rs1h	IVI	N	N	Y	N	Y
Janssen2008 [80]	mouse	*Rs1h* ^−/y^	AAV5-mOP-hRS1	SRI	N	N	Y	N	Y
Takada2008 [81]	mouse	*Rs1h*-KO	AAV2-CMV-Rs1h	IVI	N	N	Y	N	Y
Park2009 [82]	mouse	*Rs1h*-KO	AAV2-CMV-Rs1h;	IVI	N	N	Y	N	Y
			AAV8-CMV-hRSp4						
Byrne2014 [54]	mouse	*Rs1h* ^−/y^	AAV7m8-shH10-CAG-hRS1;	IVI	N	Y	Y	N	Y
			AAV7m8-rho-hRS1;						
			AAV7m8-CAG-hRS1						
Ou2015 [83]	mouse	*Rs1h*-KO	AAV8-scRS/IRBP-hRS	IVI	N	N	Y	N	Y
Ye2015 [84]	mouse	*Rs1h* ^−/y^	rAAV2tYF-CB-hRS1	IVI	N	N	N	N	N
Bush2016 [85]	mouse	*Rs1h*-KO	AAV8-scRS/IRBP-hRS	IVI	N	Y	Y	N	Y
Marangoni2016 [55]	mouse	*Rs1h*-KO	AAV8-scRS/IRBP-hRS	IVI	N	N	Y	N	N
Zeng2016 [86]	mouse	*Rs1h*-KO	AAV8-scRS/IRBP-hRS	IVI	N	Y	N	N	N
Marangoni2017 [87]	mouse	*Rs1h*-KO	AAV8-scRS/IRBP-hRS	IVI	N	N	Y	N	N
Vijayasarathy2021 [17]	mouse	*Rs1h*-KO	AAV8-scRS/IRBP-hRS	SRI	N	Y	N	N	Y
Zeng2022 [75]	rat	*Rs1h*^−/y^ rat (1)	AAV8-scRS/IRBP-hRS	IVI	N	Y	N	N	N
Ye2022 [76]	rat	*Rs1h*^−/y^ rat (2)	AAV8-scRS/IRBP-hRS	IVI	N	Y	N	N	Y
Scruggs2022 [88]	mouse	*Rs1h*-KO	AAV2/4-CMV-RS1	SRI and IVI	N	Y	Y	Y	Y
Vijayasarathy2022 [34]	mouse	*Rs1h*-KO	rAAV8-Ple155-RS1;	SRI	N	Y	Y	N	Y
			rAAV2/2-Y444-miniGluR6-RS1	IVI		N	N		

**Table 3 ijms-25-01267-t003:** The studies that reported OCT data after adeno-associated viral vector (AAV)-mediated gene augmentation therapy in Rs1-deficient mouse models. The studies are divided according to their experimental groups, and identified as First Author Publication Year, followed by a short identifier to distinguish multiple experimental groups. In one case, OCT images were obtained 15 months post-injection, by which time cavities in untreated control mice will have disappeared. Thus, this parameter is non-applicable (N/A). 1E6–2.5E9 = titers ranging from 10^6^ to 2.5 × 10^9^; d = days post-injection; shH10 = AAV subtype derived from AAV6; CAG = promoter element consisting of cytomegalovirus early enhancer; rabbit beta-globin splice adapter, and the promoter, first intron and exon of chicken beta-actin; rho = rhodopsin; IVI = intravitreal injection; SRI = subretinal injection; = a human-DNA MiniPromoter specific to ON bipolar cells; d = days; w = weeks; m = months; N/A = not applicable; NR = not reported.

Experiment ID	Timing (Post-Injection)	Cavities (Size and Amount)	Retinal Organization and Lamination	Retinal Thickness
Bush2016_1E6 [85]	14 ± 2 w	No effect	Not reported	No effect
Bush2016_1E7 [85]	14 ± 2 w	No effect	Not reported	No effect
Bush2016_5E7 [85]	14 ± 2 w	No effect	Not reported	No effect
Bush2016_1E8 [85]	14 ± 2 w	Reduced	Not reported	Increased
Bush2016_5E8 [85]	14 ± 2 w	Reduced	Not reported	Increased
Bush2016_2.5E9 [85]	14 ± 2 w	Reduced	Improved	Increased
Zeng2016 [86]	12–15 w	Reduced	Improved	Increased
Vijayasarathy2021 [17]	35 d	Reduced	Improved	Increased
Zeng2022 [75]	12 w	Absent	Improved	Increased
Ye2022_10dpi [76]	10 d	Reduced or absent	Improved	NR
Ye2022_25dpi [76]	25 d	Absent	Improved	NR
Byrne2014_shH10 [54]	4 m	No effect	No effect	No effect
Byrne2014_CAG [54]	4 m	Reduced	Improved	Increased
Byrne2014_rho [54]	4 m	Reduced	Improved	Increased
Byrne2014_15m [54]	15 m	N/A	Improved	Increased
Scruggs2022_IVI [88]	2 w	Reduced or absent	Improved	NR
Scruggs2022_SRI [88]	2 w	Reduced or absent	Improved	NR
Vijayasarathy2022_Ple155 [34]	5 w	Similar or reduced	Improved	NR

**Table 4 ijms-25-01267-t004:** A list of outcomes following adeno-associated (AAV)-mediated gene augmentation therapy, assessed using immunohistochemistry (IHC). The studies were assigned a study ID formatted as First-Author Publication year. All studies that reported IHC data following AAV-mediated gene augmentation therapy showed immunostaining for the delivered transgene. Lamination, ONL cell counts, microglial and Müller cell alterations were all addressed in two papers each. ONL = outer nuclear layer, d = days post-injection, w = weeks post-injection, m = months post-injection.

Outcomes Assessed	Timing Median <Range>	Number of Publications	Study IDs
Transgene expression	11 w <1 w, 15 m>	13 (25 experimental groups)	Min2005 [77], Molday2006 [78], Kjellstrom2007 [79], Janssen2008 [80], Takada2008 [81], Park2009 [82], Byrne2014 [54], Ou2015 [83], Bush2016 [85], Vijayasarathy2021 [17], Scruggs2022 [75], Ye2022 [76], Vijayasarathy2022 [34]
Laminar organization and integrity	32 w <25 d–15 m>	2 (4 experimental groups)	Byrne2014 [54], Ye2022 [76]
ONL cell count	4 w <1 w–15 m>	2 (7 experimental groups)	Janssen2008 [80], Byrne2014 [54]
Presence of cavities	5 w, 2 m and 27 w	2 (3 experimental groups)	Ou2015 [83], Vijayasarathy2022 [34]
Microglial/Muller cell alterations	25 d and 35 d	2	Vijayasarathy2021 [17], Ye2022 [76]
Synaptic pathology	32 w <2 m, 15 m>	3 (5 experimental groups)	Takada2008 [81], Byrne2014 [54], Ou2015 [83]

**Table 5 ijms-25-01267-t005:** Study characteristics of alternative delivery and non-viral methods to integrate and edit *RS1* tested in vitro. Characteristics regarding the study design, cell model used for the optimization, vector composition, and promotor were extracted. Studies using different strategies were divided into individual experimental groups. RS1 = retinoschisin, SLN = solid lipid nanoparticle, exo-AAV = exosome-associated AAV, cNDs = carboxylated nanodiamonds, CRISPR = Clustered Regularly Interspaced Palindromic Repeats, SMNP = supramolecular nanoparticles, C = cytosine, T = thymine, KO = knock-out, KI = knock-in, CEP4 = vector encompassing the CMV promoter and a hygromycin resistance marker, CMV = cytomegalovirus, CAG = complex consisting of CMV early enhancer (C), chicken beta-actin promoter, first intron and first exon (A), and rabbit beta-globin splice adaptor (G); GFP = green fluorescent protein, Zsgreen = *Zoanthus* sp. green, sgRNA = single guide RNA, AAV = adeno-associated viral vector, U6 = type III RNA polymerase promoter, pRS1HR-gRNA = plasmid including a 950-bp RS1 repair template and the gRNA expression cassette.

Study ID [Ref.]	Study Design	Cell Model	Plasmid
Delgado2012 [95]	*RS1* delivery in SLNs	ARPE-19 cells	CEP4-RS1
Apaolaza2015 [63]	*RS1* delivery in SLNs	ARPE-19 cells	pCAG-GFP_CMV-RS1
Wang2021 [93]	*RS1* delivery in exo-AAV	HEK-293T cells/ARPE-19/Fibroblast cells	exo-AAV2-RS1-ZsGreen
Yang2020 [97]	cND-mediated CRISPR-Cas9 KO. c.625 C > T mutation	Human iPSCs	Cas9-GFP and RS1-sgRNA
Chou2020 [96]	Dual SMNP CRISPR/Cas9-mediated KI	B16 mouse melanoma cell line	Cas9/sgRNA-plasmid Donor-RS1/GFP-plasmid
Huang2019 [58]	CRISPR/Cas9 correction of disease-associated c.625 C > T mutation	Human iPSC differentiation into retinal organoids	pCas9_GFP and pRS1HR-gRNA/ pCas9_GFP, pgRNA and RS1 Oligo
	Base editing correction of disease-associated c.625 C > T mutation	Human iPSC differentiation into retinal organoids	pCas9_GFP and pRS1HR-gRNA/ ABE7.10

**Table 6 ijms-25-01267-t006:** Study characteristics of non-viral methods evaluated in rodents. Characteristics regarding the species and strain used, delivery methods, strategy, administration route, follow-up timepoints after injection, persisting GFP expression over time, and controls were extracted. SLN = solid lipid nanoparticle, exo-AAV = exosome-associated adeno-associated viral vector, cND = carboxylated nanodiamond, SMNP = supramolecular nanoparticle, KO = knock-out, C = cytosine, T = thymine, KI = knock-in, GFP = green fluorescent protein, RS1 = retinoschisin, Rs1h = rodent homolog for RS1, IVI = intravitreal injection, SRI = subretinal injection, h = hours, w = weeks, m = months, d = days, NR = not reported.

Study ID	Model	Strain	Delivery Method	Strategy	Administr-Ation Route	Injected Volume (µL)	Interven-Tion Age	Follow-Up	Persisting GFP Expression over Time
Apaolaza2015 [63]	*Rs1h*^−/y^ mouse	C57BL/6	SLN	*RS1* integration	IVI, SRI	0.75 μL	14 w	1 w, 2 w, and 2 m	NR
Wang2021 [93]	WT mouse	C57BL/6	exo-AAV2	*RS1* integration	IVI	1 μL	4–5 w	2 w	NR
Yang2020 [97]	WT mouse	C57BL/6	cNDs	*RS1* editing (KO (c.625C > T))	IVI	5 μL	6–10 w	2 w	12 days
Chou2020 [96]	WT mouse	BALB/c	Dual SMNP	*RS1* editing (KI)	IVI	1 μL	NR	2 w	30 days

**Table 7 ijms-25-01267-t007:** List of readouts performed in in vitro studies using *RS1* integration strategy to asses transduction efficiency. The following readouts were used in order to evaluate transfection efficiency and retinoschisin (*RS1)* integration: % GFP-positive cells, mRNA and protein expression levels, cellular uptake, and secreted RS1. GFP = green fluorescent protein, h = hours.

Outcomes Assessed (and Technique)	Evaluation Time Point (Post-Transfection)	Number of Publications	Study IDs
In vitro transfection capacity (quantification of GFP and RS1)	72 h	2	Delgado2012 [95], Apaolaza2015 [63]
GFP and RS1 visualization	72 h	2	Delgado2012 [95], Apaolaza2015 [63]
In vitro transfection capacity (GFP detection)	72 h	1	Wang2021 [93]
In vitro transfection capacity (mRNA and protein expression)	48 h and 72 h	1	Wang2021 [93]
Cellular uptake (Flow cytometry)	2 h	2	Delgado2012 [95], Apaolaza2015 [63]
Cell viability (cell counting kit-8)	72 h	2	Delgado2012 [95], Apaolaza2015 [63]

**Table 8 ijms-25-01267-t008:** List of readouts performed in in vitro studies using retinoschisin (*RS1)* editing strategy to evaluate correction efficiency. All studies that edited *RS1* evaluated the transfection efficiency and the *RS1* editing using the following readouts: % GFP-positive cells, GFP and RS1 visualization, cell viability, PCR and Sanger sequencing. GFP = green fluorescent protein, PCR = polymerase chain reaction, qPCR = quantitative PCR, ddPCR = digital droplet PCR, h = hours, d = days, NR = not reported.

Outcomes Assessed (and Technique)	Evaluation Time Point (Post-Transfection)	Number of Publications	Study IDs
In vitro transfection capacity (% GFP-positive cells)	21 d	1	Chou2020 [96]
RS1 visualization	48 h	1	Chou2020 [96]
GFP visualization	2, 3, 5, 7, 14 and 21 d	1	Chou2020 [96]
	NR	1	Yang2020 [97]
Cell viability (cell counting kit-8)	24 h	1	Chou2020 [96]
	48 h	2	Yang2020 [97], Chou2020 [96]
*RS1* editing (qPCR, ddPCR)	48 h (ddPCR)	1	Yang2020 [97]
	21 d (qPCR)	1	Chou2020 [96]
	NR	1	Huang2019 [58]
*RS1* editing (Sanger sequencing)	21 d	1	Chou2020 [96]
	NR	1	Huang2019 [58]

**Table 9 ijms-25-01267-t009:** List of readouts performed in in vivo studies using retinoschisin (*RS1*) integration strategy. The following readouts were used in order to evaluate transfection efficiency and *RS1* integration: GFP and RS1 visualization, structural organization, PR loss, presence of schisis, retinal thickness, ONL thickness, mRNA and protein expression levels. GFP = green fluorescent protein, IHC = immunohistochemistry, PR = photoreceptors, ONL = outer nuclear layer, w = weeks, m = months, NR = not reported.

Outcomes Assessed (and Technique)	Evaluation Time Point (Post-Transfection)	Number of Publications	Study IDs
GFP and RS1 visualization (IHC)	1 w, 2 m	1	Apaolaza2015 [63]
	2 w	1	Wang2021 [93]
Layer organization	2 w, 2 m	1	Apaolaza2015 [63]
PR loss	2 w, 2 m	1	Apaolaza2015 [63]
Schisis (gaps)	2 w, 2 m	1	Apaolaza2015 [63]
Retinal thickness (μm)	2 w, 2 m	1	Apaolaza2015 [63]
ONL thickness (μm)	2 w, 2 m	1	Apaolaza2015 [63]
In vivo transfection capacity (mRNA and protein expression)	2 w	1	Wang2021 [93]

**Table 10 ijms-25-01267-t010:** List of readouts performed in in vivo studies using retinoschisin (*RS1*) editing strategy. The following readouts were used in order to evaluate transfection efficiency and *RS1* editing: GFP and RS1 visualization, retinal structure, genome editing efficiency, retinal thickness, inner/outer segments (IS/OS) thickness. GFP = green fluorescent protein, IHC = immunohistochemistry, OCT = optical coherence tomography, SLO = scanning laser ophthalmoscopy, H&E = haematoxylin and eosin, PR = photoreceptors, ONL = outer nuclear layer, w = weeks, m = months, NR = not reported.

Outcomes Assessed (and Technique)	Evaluation Time Point (Post-Transfection)	Number of Publications	Study IDs
GFP and RS1 visualization (IHC)	2 w	2	Yang2020 [97], Chou2020 [96]
GFP visualization (OCT)	2 w	1	Chou2020 [96]
GFP visualization (SLO)	2 w	1	Yang2020 [97]
Genome editing efficiency	2 w	2	Yang2020 [97], Chou2020 [96]
Retinal thickness (μm)	2 w	1	Yang2020 [97]
IS/OS thickness (μm)	2 w	1	Yang2020 [97]
Retinal structure (OCT, H&E)	2 w	1	Chou2020 [96]

## Data Availability

Data are contained within the article and Appendix A.

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
