# Peer review of "The Road towards Gene Therapy for X-Linked Juvenile Retinoschisis: A Systematic Review of Preclinical Gene Therapy in Cell-Based and Rodent Models of XLRS"

_ijms, 2024, doi:10.3390/ijms25021267_

Round 1
Reviewer 1 Report
Comments and Suggestions for Authors
The current manuscript is an interesting and well-done systematic review with metanalysis on gene therapy for X-linked juvenile retinoschisis. It is overall complete and extensive. Hence, I only advise that the following alterations be made before acceptance for publication:
- The academic degrees of the authors should be removed from the author list, since usually only their names should appear;
- If this is in fact a systematic review, then the article type should be changed to “Systematic Review” (on the front page);
- Table 1 should be inserted as a table, and not as a figure (as it currently is);
- More should be added on formulation composition for each analyzed study, not just for some sections (included for example on section “5.2.2 Non-viral chemical methods”, but not section “5.3.1 Non-viral physical method”; was a solution administered, or was it a suspension, or another formulation? What was its composition, water, ethanol, another solvent, were there any other excipients present (preservatives, stabilizers, permeation enhancers, etc.)?;
- Administration routes (performed, in case of in vivo studies, and intended, in case on in vitro studies) should also be mentioned in each included article;
- In Figure 11, images of the mentioned nanoparticles should be added (instead of just circles with different sizes).
Reviewer 2 Report
Comments and Suggestions for Authors
This is a very complete review mainly on gene therapy for X-linked juvenile
retinoschisis (systematic review of preclinical gene therapy in cell-based and rodent models of XLRS).
The review is outstanding under its present form. The methodology is appropriated, it is well presented and illustrated. The different parts of the manuscript are clear. The parameters which can have an impact on gene therapy (age, immunity) are taken in consideration. Alternative non viral strategies are also considered.
Whereas this is an excellent review under its present form, I think that additionnal data could be presented on patents available in the field. Are there patents for viral and non viral gene therapy, can we expect treatments in humans?
With this additionnal information, I think that this review can constitute a reference in the field.
